# Bioprinting microporous functional living materials from protein-based core-shell microgels

Yangteng Ou [1,2,3], Shixiang Cao[4], Yang Zhang[1], Hongjia Zhu[2], Chengzhi Guo[1,7], Wei Yan[4], Fengxue Xin[4], Weiliang Dong[4], Yanli Zhang[2], Masashi Narita [5], Ziyi Yu[1] ✉ & Tuomas P. J. Knowles [2,6] ✉

Living materials bring together material science and biology to allow the engineering and augmenting of living systems with novel functionalities. Bioprinting promises accurate control over the formation of such complex materials through programmable deposition of cells in soft materials, but current approaches had limited success in fine-tuning cell microenvironments while generating robust macroscopic morphologies. Here, we address this challenge through the use of core-shell microgel ink to decouple cell micro-environments from the structural shell for further processing. Cells are microfluidically immobilized in the viscous core that can promote the formation of both microbial populations and mammalian cellular spheroids, followed by interparticle annealing to give covalently stabilized functional scaffolds with controlled microporosity. The results show that the core-shell strategy mitigates cell leakage while affording a favorable environment for cell culture. Furthermore, we demonstrate that different microbial consortia can be printed into scaffolds for a range of applications. By compartmentalizing microbial consortia in separate microgels, the collective bioprocessing capability of the scaffold is significantly enhanced, shedding light on strategies to augment living materials with bioprocessing capabilities.

Living materials are complex materials that incorporate living cells into non-living components[1,2]. Depending on the nature of interactions with the component cells, such materials can range from bioinert media that offer scaffolding functions[3–7], to cell-instructive biomaterials that can direct cell behaviors[8–10], and recently, even to genetically programmable matrices produced by cells that mimic the natural formation of biofilms[11–15]. The functionality of such composite materials mostly stems from embedded cells; as such, the material should always cater for cells to grow and perform properly. However, there are typically also strong constraints on the macroscopic material properties due to the functional requirement for a final structure to present a physical form that can be handled, delivered, preserved, reused, and is protective of cells. The synergy between materials and biology has not only dramatically transformed our understanding of cellular processes[16], but also brought

[1]State Key Laboratory of Materials-oriented Chemical Engineering, College of Chemical Engineering, Nanjing Tech University, 30 Puzhu South Road, Nanjing 211816, P. R. China. [2]Yusuf Hamied Department of Chemistry, University of Cambridge, Lensfield Road, Cambridge CB2 1EW, UK. [3]Cambridge University-Nanjing Centre of Technology and Innovation, 126 Dingshan Street, Nanjing 210046, P. R. China. [4]State Key Laboratory of Materials-oriented Chemical Engineering, College of Biotechnology and Pharmaceutical Engineering, Nanjing Tech University, 30 Puzhu South Road, Nanjing 211816, P. R. China. [5]Cancer Research UK Cambridge Institute, University of Cambridge, Cambridge, Li Ka Shing Centre, Robinson Way, Cambridge CB2 0RE, UK. [6]Cavendish Laboratory, University of Cambridge, J J Thomson Avenue, Cambridge CB3 0HE, UK. [7]Present address: Department of Chemical Engineering, University College London, Torrington Place, London, WC1E 7JE, UK. ✉e-mail: ziyi.yu@njtech.edu.cn; tpjk2@cam.ac.uk

us the capacity to engineer living systems towards a myriad of applications, from therapeutical delivery of cells for regenerative medicine[17–19], to on-demand production of high-value chemicals through microbial bioprocessing[6,7].

Manipulating the spatial distribution of cells is one of the most sought-after capabilities in the field of living materials. Bioprinting has arguably captured the most attention[20,21] on account of its versatility and compatibility with many cell-friendly soft materials. For instance, bioprinting enables the programmable deposition of mammalian cells in bioactive hydrogels to create 3D biological constructs that better recapitulate the intricacy and heterogeneity of native tissues[22–25], which holds tremendous translational values in biomedicine. Bioprinting microbes have also seen increased applications in recent years, as it furthers our understanding of dynamic bacterial communities[26] and affords insights for bioprocessing enhancement[3–6]. However, current bioprinting routines often trade off suitability to cells for improved material manufacturability, as the inherent mechanical and rheological properties of the bioink cannot be properly decoupled from cellular microenvironments[27,28]. Moreover, it remains a challenge to construct arbitrary macroscopic material morphologies with well-defined cell niches to establish robust interactions among different cell communities[20,21,29].

Here, we propose a method to address this challenge. In this work, we explore the applicability of cell-laden core-shell microgels at the interface of bioprinting and functional living materials. Based on an aqueous two-phase system between gelatin/gelatin methacryloyl (gelMA) and carboxymethylcellulose (CMC)[30], core-shell microgels are fabricated with cells encapsulated in the viscous core phase, while the hydrogel shell affords a dual-networking strategy to be covalently bonded with each other into a microporous PAM (Protein-based Annealed Microgel) scaffold. Furthermore, we bridge microfluidically tunable building blocks with functionalities of macroscopic living materials via extrusion bioprinting[29,31] towards bioprocessing. We show that core-shell microgels support the growth of both microbial populations and mammalian cellular spheroids; more importantly, in comparison with their non-core-shell counterparts, the core-shell microgels can reduce the cell leakage to the medium while augmenting the bioprocessing capacity of such materials. Lastly, we demonstrate that by fabricating scaffolds with locally varying properties, i.e., heterogeneously distributed and spatially segregated cell populations in discrete microgels, significantly enhanced bioactivities are observed in two typical microbial consortia models. We believe our method has provided a generalizable strategy to construct the next generation of living materials, holding promises not only in microbial bioprocessing, but also in advanced biofabrication.

## Results and discussions
### From core-shell building blocks to microporous PAM scaffolds
We started with gelatin, a collagen-derived material that has long been used in a variety of biological applications on account of its animal origin and wide availability[32], to construct living materials (Fig. 1). To afford a dual-network, gelatin was first blended with its photo-crosslinkable derivative, gelMA, to generate a homogenous 15% hydrogel precursor solution spiked with 0.5% blue light photoinitiators lithium phenyl-2,4,6-trimethylbenzoylphosphinate (LAP). GelMA was synthesized according to a well-established protocol[33], with an estimated degree of functionalization of 50% by NMR (Supplementary Fig. 1). The dispersed phase was emulsified in a microfluidic flow-focusing device, where a solution of 1% carboxymethylcellulose (CMC) spiked with 10 U/ml transglutaminases was flanked by hydrogel-forming solutions at the device junction, and together the dispersed phase was intersected by the carrier oil (Novec 7500 with 0.1% Pico-surf surfactant) to generate droplets (Supplementary Fig. 2). Flow rates for hydrogel precursor, CMC, and oil were respectively 8, 2, 40 µL/min, resulting in highly monodispersive core-shell droplets of ~165 µm in

diameter (Coefficient of Variance, CV = 2.2%) and CMC core ~90 µm (CV = 7.4%). By microfluidics, the shell thickness could be controlled (Fig. 1b). Droplets were cured at room temperature overnight to allow the diffusion of transglutaminases from the CMC core into the shell phase which, subsequently, catalyzed the formation of isopeptide bonds between glutamine and lysine and hence the first covalent network. Microgels were demulsified, followed by a direct mixing with phosphate buffered solution (PBS) containing 0.5% LAP at 4:1 volume ratio to yield a satisfactory printability. Using an extrusion 3D printer, jammed microgels could be patterned into macroscopic scaffolds. Finally, a 405 nm blue light initiated a second crosslinking among microgels to generate covalently stabilized PAM scaffolds mechanically stiff enough to be readily handled by a pair of tweezers (Fig. 1c) and can maintain its structural integrity after 3-day incubation in PBS (Supplementary Video 1). Noticeably, the dual covalent strategy is reversible, i.e., droplets can be first cured by blue light radiation and then annealed enzymatically.

SEM (Scanning Electron Microscope) images show the core-shell morphology of the building blocks (Supplementary Fig. 3b) and that the dual-networked PAM scaffolds bore multi-scale porosity (Supplementary Fig. 3a–c). At microgel surface there were nanopores (Supplementary Fig. 3a) which are characteristic of hydrogels as a high-water-content material. Microgel packing gave rise to micropores[34], as evidenced by the stretched mesh morphology of lyophilized PAM scaffolds (Supplementary Fig. 3c). To further investigate how microgel size affects the physical properties of scaffolds at macroscale, we studied the degradation kinetics of PAM scaffolds assembled from differently sized microgels by percolating trypsin solution into the structure (Supplementary Fig. 4a), along with dual-crosslinked bulk hydrogels and non-annealed microgels. Kinetics profiles (Supplementary Fig. 4b) demonstrate that the degradation of all PAM scaffolds, regardless of the microgel sizes, showed higher rates than bulk hydrogels owing to the presence of micropores that facilitated percolation of the trypsin solution into the scaffolds. Moreover, the degradation tended to be slower with smaller microgel sizes possibly due to decreased pore sizes[19] and hence slowed percolation (Supplementary Fig. 4b).

### Direct writing of microgel inks via extrusion printing
Microgels have recently emerged as a type of ink to fabricate 3D structures through extrusion printing[29,31,34–37]. In a jammed state, they can form rest structures behaving as elastic solids through physical interactions such as frictions; when sheared, the interparticle frictions are dissipated to allow the ink to flow. As such, materials that can be fashioned into microgels are theoretically printable via extrusion irrespective of their polymeric chemistry[29,31]. To quantitatively evaluate the printability of this core-shell microgel ink, rheology characterization was carried out first. Jammed microgels exhibited shear-thinning behavior (Fig. 2a) and could quickly recover from strain upon its removal (Fig. 2b). Collectively, this particular rheological behavior can be translated into the ability of the ink to rapidly transform from flowing liquid when sheared, to elastic solid upon leaving the nozzle. Strain sweep experiment demonstrates that the microgel ink yielded at around 30% strain (Supplementary Fig. 5a); after interparticle annealing, both the storage and loss modulus increased significantly (Fig. 2c) and the yield stress witnessed a roughly sixfold increase (Supplementary Fig. 5b), giving rise to PAM scaffolds with enhanced mechanical strength. Moreover, the ink with a reversed dual-networking strategy shared a set of similar rheological properties (Supplementary Fig. 5c–f).

We next tested the fidelity of printing. Nozzles of differing inner diameters were employed to pattern scaffolds, and the filaments all displayed uniform thickness without noticeable flattening (Fig. 2d) otherwise affected by surface tension upon contacting with the solid surface. Consequently, the filament diameters were faithful to printing nozzles and even slightly smaller using 20 G, 21 G, and 22 G nozzles

(Fig. 2e) owing to increased edge jaggedness. We then tested the compatibility of the microgel ink with a commercial extrusion 3D printer (Supplementary Fig. 6a and Supplementary Video 2). Microgel ink with optimized printability was loaded into a dispersion cartridge coupled with a three-axis mechanical robotic arm. The extrusion and the writing speed of the microgel ink were controlled pneumatically and by the robotic arm, respectively. Unlike continuous material-based extrusion printing where the filament thickness can be controlled by both the extrusion and the writing speed due to filament fusion[38], we found that the thickness of the printed microgel ink is barely depen-

dent on neither speed but the size of the nozzle due to its more elastic behavior. Thus, mismatched speeds will lead to either filament breaking (extrusion<writing) or undesired filament buildup (extrusion>writing, Supplementary Fig. 6b). As a result, by matching the speed of ink extrusion and writing, we were able to pattern the microgel ink into various predefined shapes (Supplementary Fig. 6a) and multilayered structures. Besides homogenous structures (Fig. 2fi and ii), heterogenous structures (Fig. 2f, iii, and iv) could also be fabricated by simply mixing populations of microgels carrying different properties.

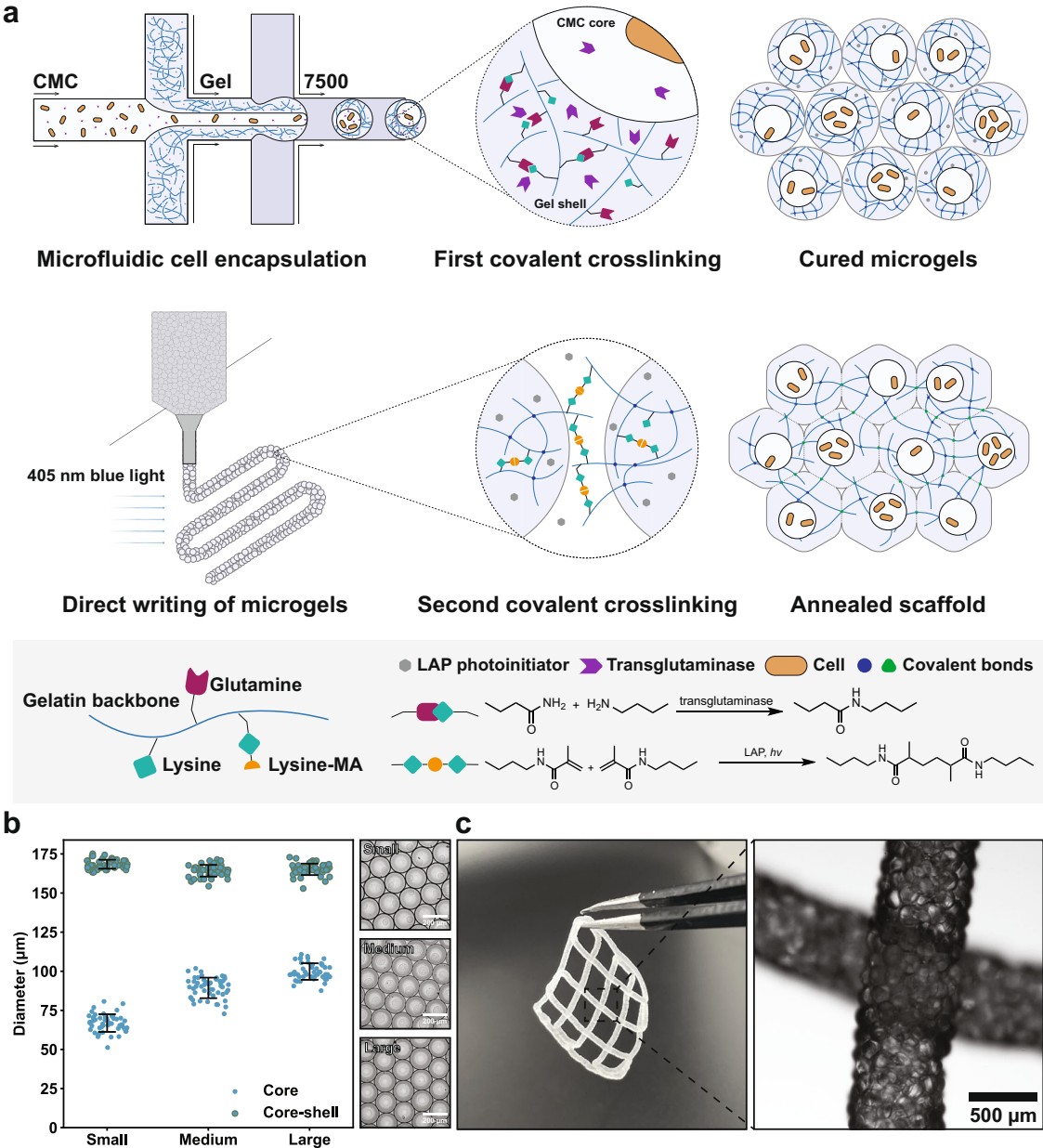

**Fig. 1 | Microfluidic templating of core-shell microgel building blocks for fabricating PAM scaffolds. a** Schematic of the formation of PAM scaffolds. First, the LAP-containing gelatin/gelMA polymer blend, together with cell-containing CMC solution spiked with transglutaminases, is emulsified by carrier oil into droplets within a flow-focusing microfluidic device. Droplets are then converted into microgels by enzymatic reactions, after which they are extruded and patterned into scaffolds. Finally, blue light initiates a second covalent crosslinking among jammed microgels, giving rise to PAM scaffolds. **b** By microfluidics, the core-shell ratio can

be controlled. Flow rates of the hydrogel precursor solution (shell phase) and the carrier oil are controlled at 8 and 40 μL/min, respectively, and the flow rates of CMC (core phase) are 1, 2, and 3 μL/min, for small, medium, and large core size, respectively, $n = 50$ microgels analyzed in one experiment. **c** Printed PAM scaffolds and the micrograph shows local magnification of two intersected filaments. Data are presented as mean values ± standard deviation and source data are provided as a Source Data file.

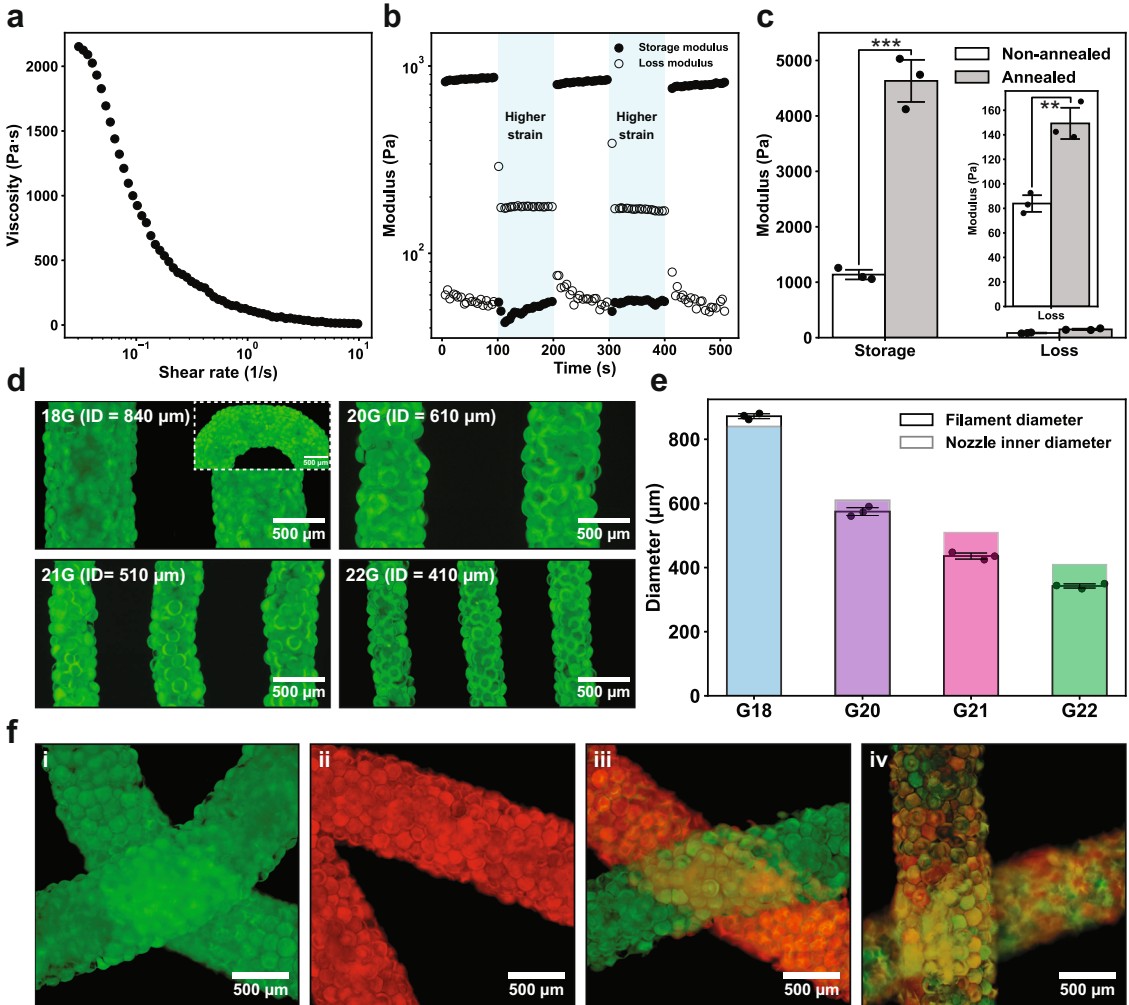

**Fig. 2 | Direct ink writing of jammed microgels via extrusion printing.**
**a**–**c** Rheology characterization of the microgel ink. Microgels are enzymatically cured. **a** Shear rate sweep experiment. Shear rate from 0 to 10 1/s. Strain 1%. **b** Step-strain sweep where low strain (1%) and high strain (90%) are cycled every 100 s. Frequency 1 Hz. **c** Storage and loss moduli before and after the photo-initiated annealing, $n = 3$ independent rheological characterizations. ***$p = 0.00022$, ** $p = 0.0031$, unpaired two-tailed student's $t$ test. **d** Fluorescent images of extruded filaments printed from differently sized nozzles. Inset shows a curved corner of a

filament. **e** Analysis of the filament diameters. Colored bars represent the inner diameters of the printing nozzles and the blank ones are the measured diameters of printed filaments, $n = 3$ independently printed filaments in one experiment. **f** Fluorescent images of scaffolds, i.e., (i) green and (ii) red homogeneous scaffolds, (iii) a macroscopically heterogenous scaffold, and (iv) a microscopically hetero-genous scaffold. **d**, **f** Fluorescent microgels are generated from a mixture of gelatin with fluorophore-labeled gelMA. Data are presented as mean values ± standard deviation and source data are provided as a Source Data file.

## Core-shell microgels for cell culture

One of the major challenges in extrusion bioprinting is how to strike a balance between the material mechanical properties, material manu-facturability, and suitability for cell culture[21,27,28], as the inherent mechanical features of the material, i.e., the rheological profile and the final stiffness of the material will not only determine its printing performance[20,39], but also more crucially, might adversely affect cel-lular behaviors[40,41]. To address this challenge, we utilize the core-shell strategy to decouple material processing (the shell phase) from cell culture (the core phase) through their phase separation[30]. The core material, CMC, has been used for cell culture and as tissue engineering scaffolds[42,43]. The rheology characterization of the core material shows that the CMC solution leans towards viscosity (Supplementary Fig. 5g, i). Next, we investigated the proliferation of microorganisms in such viscous environment as opposed to non-core-shell building blocks (Fig. 3). *E. coli* were allowed to grow in cured microgels for 24 hours. In core-shell microgels, to a great extent, cell growth was physically confined, which led to a concentrated bacterial population in the core (Fig. 3a). In stark contrast, bacteria in non-core-shell microgels

proliferated locally and formed rounded and sporadic microcolonies (Fig. 3b) that were similar to those reported recently[44]. Moreover, we also visually discovered that such spatial confinement could restrain bacteria from escaping to the environment more effectively (Supple-mentary Fig. 7a). We further confirmed the result via a dilution plating experiment (Supplementary Fig. 7b), which displays that the colony-forming unit (CFU) of the supernatant of non-core-shell microgels was two orders of magnitude higher than that of core-shell microgels over 24 hours (Supplementary Fig. 7c). As cell leakage constitutes an unaddressed challenge in the field of living materials for bioprocessing[6,7,45], especially when genetically modified microorgan-isms are involved that cannot risk escaping, the use of core-shell microgels might offer strategies into solutions to this problem[46]. Fur-thermore, microbes could proliferate in the PAM scaffolds in a similar fashion without notable cross-contamination among adjacent micro-gels (Supplementary Fig. 8).

Applicability of core-shell microgels for mammalian cell culture was also characterized (Fig. 3c–f). The delivery of cells into the microgels followed Poisson distribution (Supplementary Fig. 9).

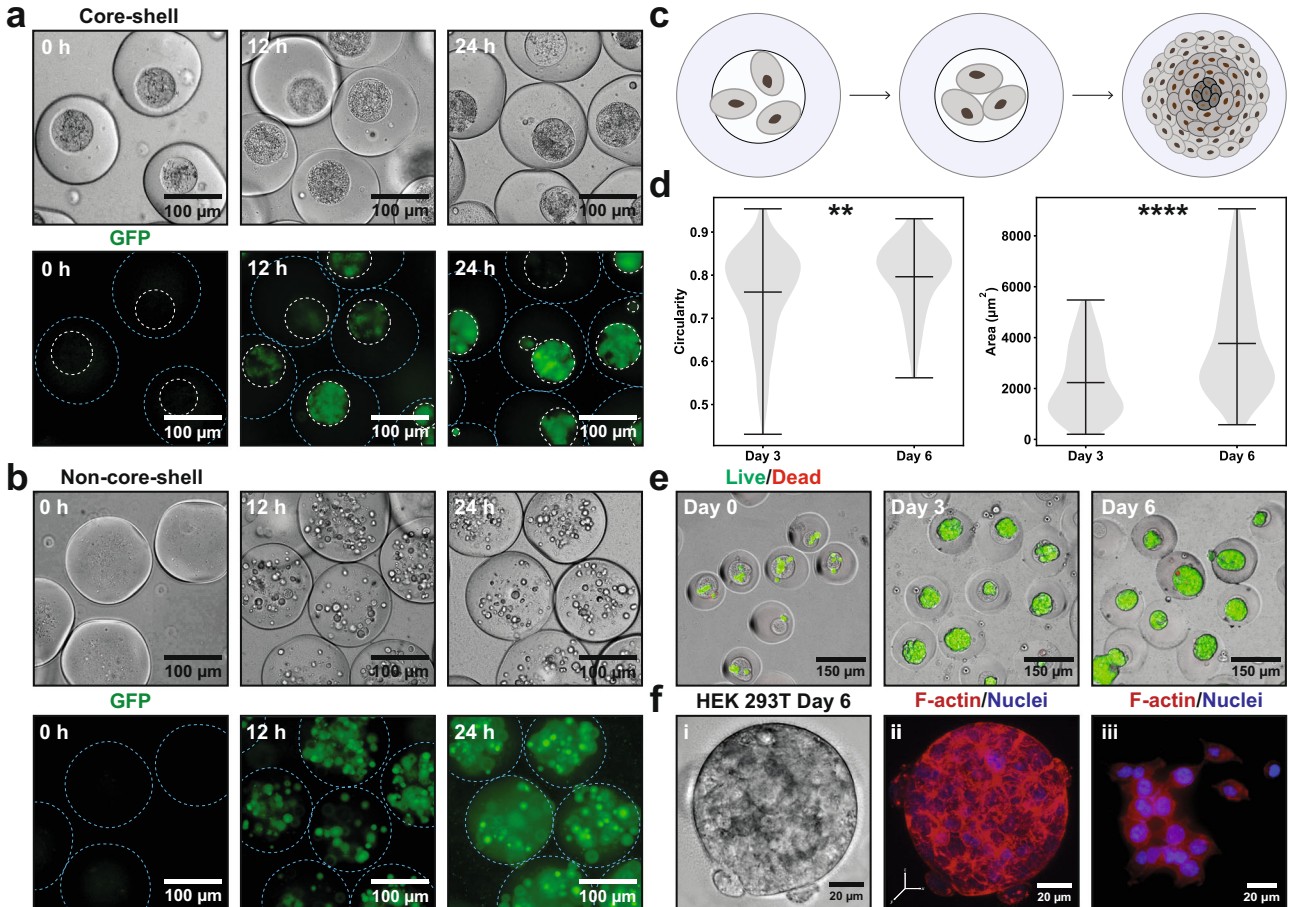

**Fig. 3 | Core-shell microgels for cell culture.** Left panel: GFP *Escherichia Coli* (*E. coli.*) in core-shell and non-core-shell microgels. Right panel: formation and characterization of HEK 293 T cellular spheroids in core-shell microgels. **a** Proliferation of GFP *E. coli.* in the core-shell microgels. Cell growth is confined inside the viscous microgel cores. **b** Proliferation of *E. coli.* in the non-core-shell microgels. *E. coli* proliferate locally to form visible microcolonies. Blue and white dashed lines outline the microgels and the CMC cores, respectively. **c** Schematic representation of the formation of multicellular spheroids. Due to spatial confinement, cells are allowed to interact in all three dimensions to form multicellular spheroids featuring a hierarchical structure. **d** Growth of HEK 293 T spheroids from day 3 to day 6, quantified by spheroid circularity and area. Data are represented as mean values with minimums/maximums, n = 107 spheroids (day 3), n = 248 spheroids (day 6) in the same experiment. \*\**p* = 0.0019, \*\*\*\**p* < 1E-15, unpaired two-tailed student's *t* test. Source data are provided as a Source Data file. **e** Live/Dead staining of cell-laden microgels at day 0, day 3, and day 6. Live cells (green) and dead cells (red) are stained with calcein acetoxymethyl (calcein AM) and ethidium homodimer-1 (EthD-1), respectively. **f** Cytoskeletal structure of HEK 293 T spheroids. (i) Brightfield image of a HEK 293 T spheroid and (ii) its confocal microscopic image. (iii) Cytoskeletal structure of HEK 293 T from monolayer culture. F-actin and nuclei are stained with phalloidin and 4', 6-diamidino-2-phenylindole (DAPI), respectively. Confocal fluorescent micrograph is rendered from cross-sectional images at different vertical positions (Supplementary Fig. 10).

Likewise, cells were spatially confined and allowed to proliferate and interact in all dimensions inside microgel cores to self-assemble into high-cell density spheroids (Fig. 3c). Human embryonic kidney (HEK) 293 T cells developed into cellular spheroids after three days in core-shell microgels (Fig. 3e), most of which displayed a high circularity (higher than 0.8, Fig. 3d). From day 3 to day 6, the size of the spheroids increased 80% (Fig. 3d) and maintained high viability, as evidenced by Live/Dead staining (Fig. 3e). Additionally, the cytoskeleton of HEK 293 T spheroids was composed of densely packed nuclei surrounded by a meshwork of cortical actin (Fig. 3f, ii and Supplementary Fig. 10), epitomizing the predominance of cell-cell interactions, in contrast to monolayer-based cell culture, where actin was organized into cytoplasmic fibers[47] (Fig. 3f, iii). To conclude, HEK 293 T are able to form well-structured cellular spheroids inside core-shell microgels with high-cell viability, indicating their potentials for future applications as the building blocks to biofabricate high-cell-density structures[20].

## Bioprinting functional living materials for bioprocessing

Printing microbes is the emerging frontier in bioprinting[48], as it offers a versatile tool to create functional living materials with well-defined shapes and properties that have found a wide range of applications including sensing[49], biomanufacturing[6], and bioremediation[3,7]. The core-shell microgels were then used as the bioink to print functional living scaffolds for bioprocessing (Fig. 4a). We began with a natural *Saccharomyces cerevisiae* fermentation system to anaerobically convert glucose into ethanol[3–5]. Yeast-laden core-shell microgels were printed (Fig. 4b) into annealed PAM scaffolds and submerged into yeast extract peptone dextrose (YPD) media in an oxygen-free environment for fermentation. For all the yeast-laden core-shell microgel scaffolds used in this work, the ethanol production started to take off 2 hours after submersion in YPD media and grew exponentially in 12 hours (Fig. 4e). First, we investigated how physical properties of the building block influence the bioprocessing capacity of printed scaffolds. By solely tuning the flow rate of the core phase, core-shell microgels of the same overall size (core-shell A and B: 156 ± 3.2 μm and 154 ± 2.0 μm, respectively) were generated with varied core sizes (84 ± 8.2 μm and 56 ± 2.1 μm, respectively, Fig. 4c, d) and thus a discrepancy in cell loadings. We probed the kinetics of ethanol production in media (Fig. 4e) and found that larger cores (core-shell A) led to a more rapid generation of ethanol because of higher cell loadings. Next,

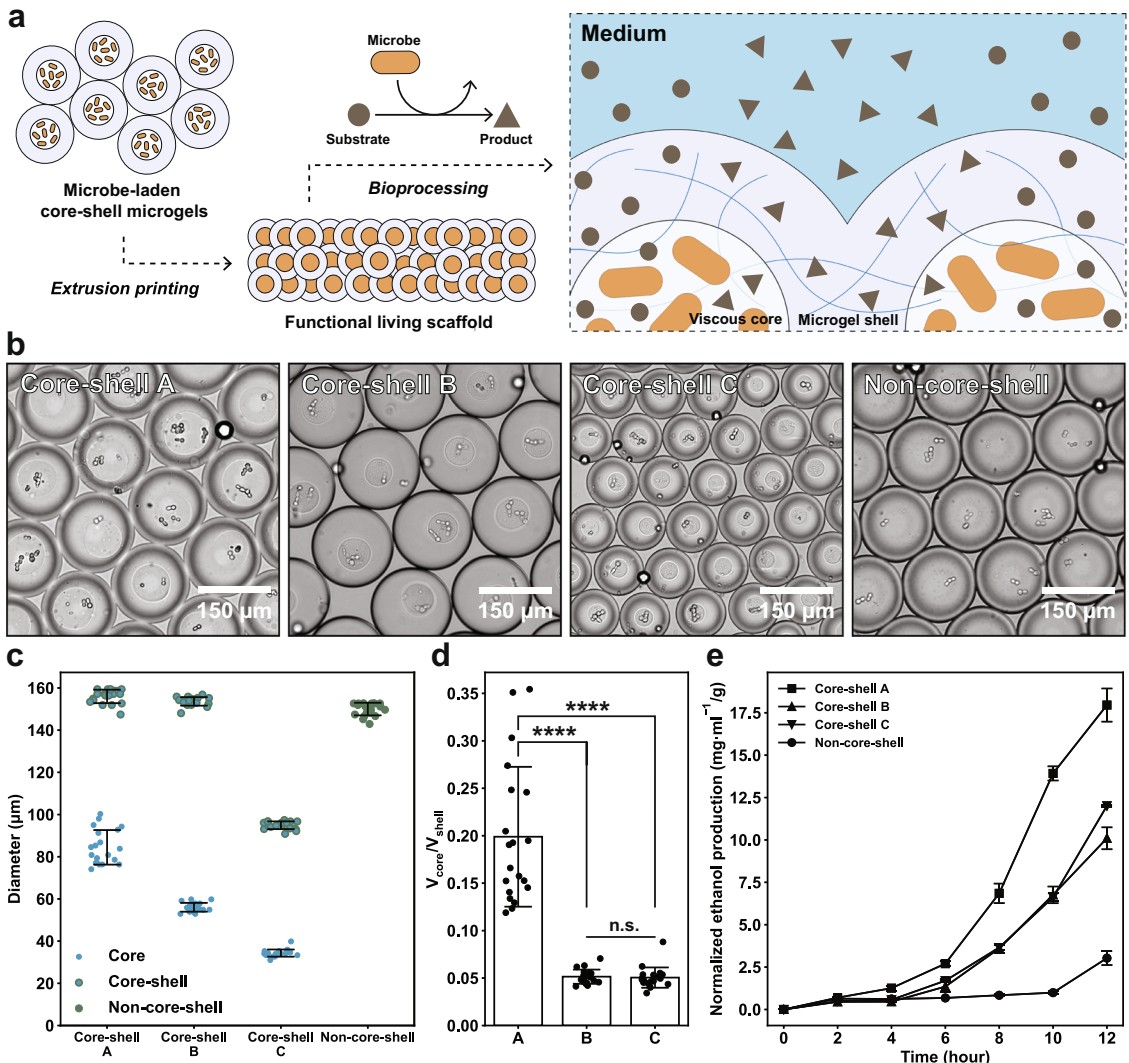

**Fig. 4 | Bioprinting yeast-laden scaffold for ethanol fermentation. a** Schematic of the fabrication of functional living scaffolds. Microbe-laden core-shell microgels are generated via droplets microfluidics and printed into annealed scaffolds that are used for bioprocessing, where the microbes act as the whole-cell biocatalyst to convert the substrate into product in the medium. **b** Micrographs of the core-shell microgels with varied sizes and non-core-shell microgels. **c** The size distribution of the core-shell microgels and non-core-shell microgels in **b**, $n = 20$ microgels in one experiment. **d** The volume ratio between the core and the shell phase for core-shell microgels. Core-shell microgels A have a similar overall size to B but have larger cores. Core-shell microgels B and C are different in overall size but they have the same core/shell ratio in order to maintain equivalent cell loadings for bioprinting, $n = 20$ microgels in one experiment, ****$p < $1E-10, n.s. (not significant) = 0.74, unpaired two-tailed student's $t$ test. **e** The production of ethanol from yeast over 12 hours in oxygen-free environment, which is normalized by the scaffold weights, $n = 3$ independent biological experiments. Data are presented as mean values ± standard deviation and source data are provided as a Source Data file.

we examined if the size of core-shell microgels would cause differences in fermentation. To dissociate the factor of cell loading from bioprocessing performance, the flow profiles of the dispersed phase (both the core and the shell) used to fabricate core-shell microgel B were kept constant but with an increased flow rate of the carrier oil in a smaller droplet microfluidic device (Supplementary Fig. 11a), giving rising to smaller core-shell microgels (core-shell C, 95 ± 3.2 μm overall, 34 ± 1.7 μm core) with an unaltered volume ratio between the core and the shell material (Fig. 4d). Therefore, by printing scaffolds of equal weights, their initial cell loadings were close regardless of the size of the building blocks. We found that with an equivalent amount of yeast cells immobilized in the scaffold, the size of the core-shell microgels did not significantly influence the fermentation process (Fig. 4e), which we ascribed to the presence of the micropores in the scaffold, along with relatively shorter diffusional length[34], that did not markedly hinder the delivery of small molecules to/from the cells. Lastly, non-core-shell microgels were also generated (150 ± 3.0 μm) for

comparison and a considerably delayed production of ethanol by scaffolds printed from thereof (Fig. 4e) was observed, mostly likely due to the restrained growth of yeast in a highly chemically crosslinked hydrogel network that had slowed down the cell proliferation[44]. However, we also found that the end-point ethanol production (20 hours) from the core-shell PAM scaffolds was slightly lower (Supplementary Fig. 11b), which we attributed to a more pronounced cell leakage for non-core-shell PAM scaffolds (Supplementary Fig. 11c) that boosted the ethanol production.

## Microgel segregation enhances microbial consortia-based bioprocessing

We next extended the application to microbial consortia-based bioprocessing[6]. Through interconnected core-shell microgels, PAM scaffolds possess not only controllable porosity but also the capability to segregate multi-species communities in separate microgels (Fig. 5a), which together effects a more efficient transfer of nutrients and

metabolites between the scaffold and environment as well as interacting species while precluding their competitive growth. We then examined whether this method could enhance bioactivities of immobilized microbial consortia through a spatial division of labor (Fig. 5b) compared to direct mixing of consortia. First, we studied a microalgae-bacteria system consisting of photoautotrophic *Chlorella vulgaris* and aerobic *Bacillus subtilis* (Fig. 5c) for its capacity to bioremediate methyl orange[3,50] and amoxicillin[51]. In this microbial consortium, the microalgae fix carbon dioxide emitted from the bacteria into carbon sources whilst producing oxygen through photosynthesis for bacteria to respire (Fig. 5c), which hence improves its bioremediation capability by reciprocally benefitting the growth of both microbes[52,53]. First, to demonstrate the bioremediation process, heterogeneous PAM scaffolds immobilizing the consortium in separate microgels were fabricated and submerged in synthetic wastewater containing methyl orange or amoxicillin (Supplementary Fig. 12a). Over 90% of amoxicillin (300 mg/L) was removed in 24 hours and around 15% for methyl orange (100 mg/L). To examine whether such spatial separation strategy augments bioremediation, homogeneous scaffolds, in which the consortium was encapsulated in the same microgels, were tested, along with scaffolds immobilizing monocultures of microalgae and bacteria, respectively. The result (Fig. 5d) demonstrates that both immobilized single microbial populations could remove methyl orange with a similar efficacy over 24 hours, but the homogenous scaffolds were more efficient in bioremediation despite of a halved cell loading for both microbes, suggesting the synergistic effect of the consortium. More importantly, the heterogeneous scaffolds remediated significantly more methyl orange (Fig. 5d), indicating an improved bioprocessing capability. However, we also observed that after 48 hours, the heterogeneous scaffolds were all degraded and liquified (Supplementary Fig. 13a), which was caused by the hydrolysis of scaffolds by microbial proteases of which the concentration increased over time in 48 hours (Supplementary Fig. 12b). We sought to utilize this phenomenon as an extra qualitative measure of microbial bioactivity of the consortium (Fig. 5e). Four PAM scaffolds of the exact arrangement as the previous were fabricated and immersed in BG11 media (Fig. 5e and Supplementary Fig. 13b). We again observed a similar pattern: while the scaffold immobilizing microalgae began to decompose after 24 hours, for 72 hours, the scaffold with bacteria largely retained its structure, as the nutrient-deficient medium is unfavorable for the growth of *Bacillus subtilis*. Within 24 hours, the heterogenous scaffold started disintegrating into smaller fragments whereas the homogenous scaffold maintained structural integrity until day 3 (Supplementary Fig. 13c).

Finally, we tested a synthetic fungi-bacteria consortium to ferment glucose into 2-phenylethanol (2-PE) (Fig. 5f). In this enzymatic cascade (Fig. 5f and Supplementary Fig. 14), the carbon source is first converted by *Escherichia Coli* into the intermediate l-phenylalanine that is further metabolized into 2-PE by *Meyerozyma guilliermondii*, a species of yeast that is reported to produce and tolerate a high concentration of 2-PE[54,60]. As both microorganisms utilize glucose as the carbon source, they form a competitive relationship. In liquid culture where two microbes are simply mixed together, the *M. guilliermondii* would always become the dominant species over 96 hours, irrespective of the initial inoculation ratio (Supplementary Fig. 15). To characterize the biomanufacturing process of 2-PE from consortium-containing scaffolds, we first measured the kinetics of 2-PE production with the heterogeneous scaffolds in synthetic media. A mixed population of microgels respectively encapsulated *E. coli* or *M. guilliermondii* were fabricated into scaffolds and incubated in the culture media to ferment glucose into 2-PE (Supplementary Fig. 16). Result displays (Fig. 5g) sigmoidal kinetics that peaked at around day 2 and gradually leveled off after 3 days of fermentation. Next, we compared the yield of 2-PE between two different sets of scaffolds (Fig. 5h).

Remarkably, over sixfold higher 2-PE concentration was detected from the fermentation broth containing heterogeneous scaffolds than their homogeneous counterparts in 3 days. We also found that there was more 2-PE from the homogeneous scaffolds in comparison with liquid culture (Fig. 5h), which was possibly attributed to the closer physical contact of two microorganisms brought by growth confinement in microgels. At last, we examined the reusability of scaffolds for 2-PE batch production (Supplementary Fig. 16). Scaffolds could maintain integrity during the first batch (3 days); despite gradual structural collapse henceforth, scaffolds were not fully compromised after 4 batch processes in a row (12 days in total). However, the production of 2-PE significantly reduced (Supplementary Fig. 16g), especially for heterogeneous scaffolds, which decreased by 70% at the second batch whereas 20% for homogeneous scaffolds. 2-PE production for both scaffolds deteriorated batch by batch and almost completely stopped after the third batch; nevertheless, the heterogeneous scaffolds still yielded more 2-PE (Supplementary Fig. 16g). The declining performance of scaffolds could be caused by cells' escaping and reattaching to the inner surface of the scaffold that upset the growth balance of two microbial species, as they could not be entirely scoured away by fresh media between batches.

Taken all together, the work presented here describes an approach for constructing living materials via a common extrusion bioprinting routine. Through a microgel dual-networking method, covalently stabilized macroscopic functional living materials can be printed and readily deployed for bioremediation and biomanufacturing. We used core-shell microgels as the building blocks and in spite of the inevitable cell leakage[6,7], we found such strategy can mitigate this problem compared to non-core-shell microgels. Additionally, using a yeast monoculture model, severely detained ethanol production was observed in non-core-shell microgel scaffolds, suggestive of a better suitability of core-shell microgels for cell culture. Furthermore, microbe-laden microgels were used to spatially organize the cell communities, from which a heterogeneous distribution of cells can be established to promote the inter-species communications of microbial consortia amid the generation of programmable morphologies by extrusion printing. Our results demonstrate remarkably improved bioactivities from two different microbial consortia. Although the protein-based material might not be the most ideal choice for microbial applications due to its susceptibility to protease hydrolysis, we envisage future efforts of using more mechanically robust materials for such applications. We also show that the core-shell microgels can culture mammalian cellular spheroids[61]; combined with the droplet microfluidics and the microgel dual-networking strategy, they hold promises to be utilized as the building blocks to construct high-cell-density structures. In conclusion, we believe our proposed method represents a valuable paradigm to construct functional living materials for microbial bioprocessing and also holds potential for advanced biofabrication[55].

## Methods

### GelMA synthesis and characterization
Gelatin methacryloyl (gelMA) was synthesized according to a published protocol[33]. Gelatin from porcine skin (Shanghai Aladdin Biochemical Technology) was dissolved in deionized water at 50 °C to a final concentration of 10% (w/v, concentrations are all denoted as w/v or stated otherwise). 0.6 g methacrylic anhydride (Shanghai Macklin Biochemical) was added to the homogenous gelatin solution dropwise per 1 g gelatin. Reaction was carried out at 50 °C for 1 h and terminated by adding two volumes of preheated deionized water, after which the mixture was centrifuged for 3 min at $3500 \times g$. The supernatants were decanted and dialyzed against deionized water at 30 °C using dialysis tubing of 12 kDa molecular weight cut-off for 7 days. After dialysis, the acidic solution of gelMA was adjusted to pH = 7.4, snap-frozen by liquid nitrogen, and lyophilized till total hydration to porous white

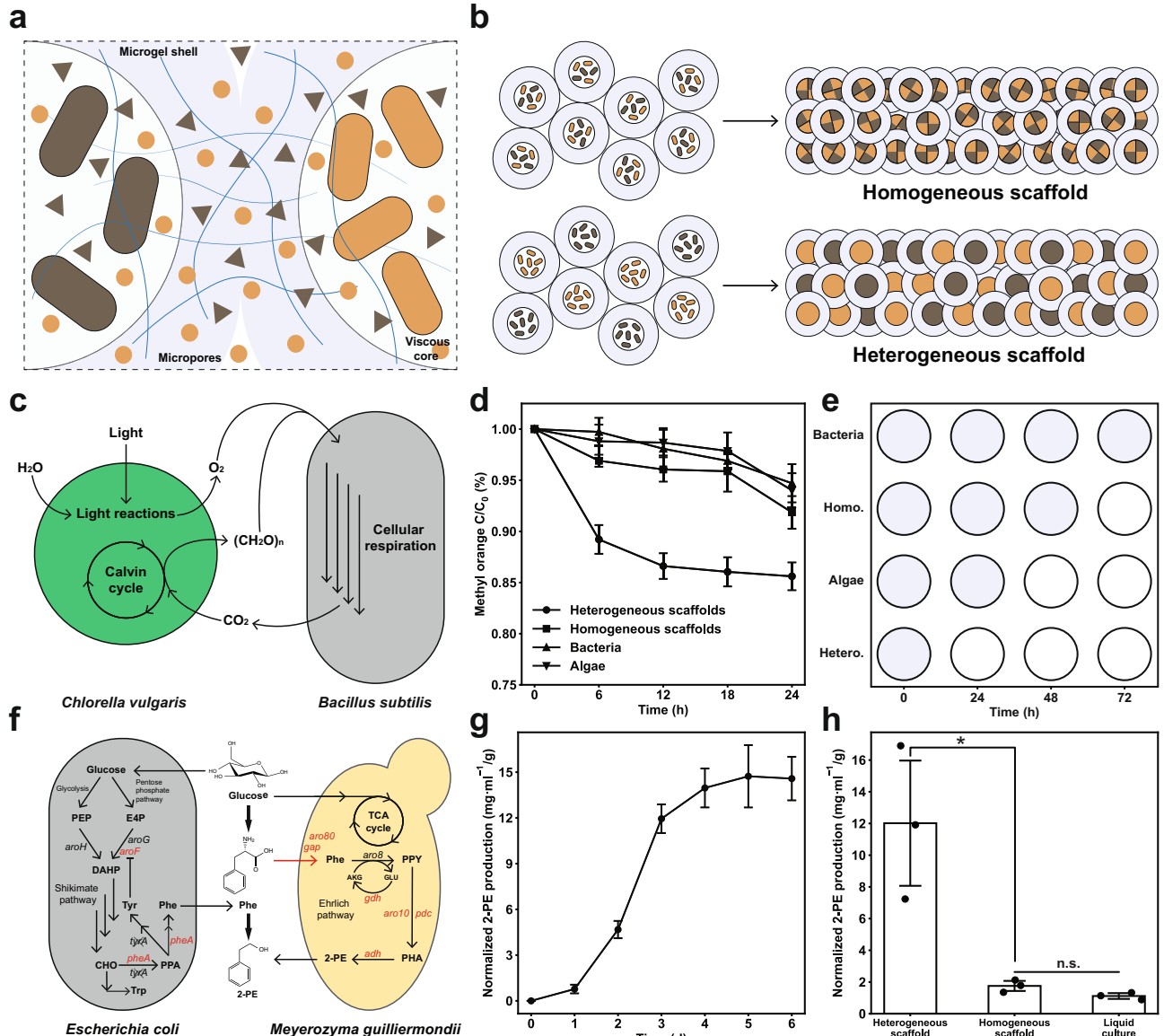

**Fig. 5 | Miscroscopic compartmentalziation enhances bioactivies of microbial consortia. a** PAM scaffolds immobolize and segregate microbial consortia at microscopic space, which prevents competetive growth while enabling a more efficient mass transfer of metabolites between speceis. Circles and triangles represent metabolites secreted by microbes. **b** Cell-laden core-shell microgels are jammed and printed into scaffolds. Heterogeneous and homogeneous scaffolds are defined as: a mixed population of microgels containing two different microorganism species, respectively, and a homogeneous population of microgels containing a mixture of two microorganisms. **c**–**e** A mutualism model of microbial consortia. **c** Schematic representation of the relationship between *Chlorella vulgaris* and *Bacillus subtilis*. **d** Bioremediation of methyl orange by heterogeneous scaffolds, homogeneous scaffolds, and scaffolds immobilizing monocultures of microalgae and bacteria, respectively, $n = 3$ independent biological experiments. **e** Characterization of scaffold disintegration in 72 hours. The closed symbol marks an unbroken scaffold whereas open symbol scaffold markedly disintegrating. For

all experiments, scaffolds are subjected to a cyclic 12:12 hour of light/darkness condition and incubated at 25 °C. **f**–**h** A competitive growth model of microbial consortia. **f** The synthetic microbial consortium of genetically engineered *Escherichia Coli* and *Meyerozyma guilliermondii* for producing 2-PE from glucose[60]. Overexpressed genes are marked red. Crossed gene *tyrA* is knocked out. **g** The production of 2-PE from heterogeneous scaffolds over 6 days, which is normalized by respective scaffold weights, $n = 3$ independent biological experiments. **h** Comparison of bioprocessing in terms of normalized 2-PE production over 3 days, $n = 3$ independent biological experiments. *$p = 0.021$, n.s. = 0.069, unpaired two-tailed student's *t* test. For both scaffolds, the production is normalized by corresponding scaffold weights; for liquid culture, CMC containing consortia is directly inoculated into media, the cell loading of which is the same as the scaffolds, and 2-PE production is normalized to average scaffolds weights of heterogeneous and homogeneous combined. Data are presented as mean values ± standard deviation and source data are provided as a Source Data file.

foam. Modification of gelatin was verified by [1]H Nuclear Magnetic Resonance (NMR). Gelatin and gelMA were dissolved in deuterium oxide, receptively, and spectra (Supplementary Fig. 1) were collected at a frequency of 400 MHz using a Bruker Avance III HD Spectrometer with a single-axis gradient inverse probe. To estimate the degree of functionalization (DoF), the integrated lysine methylene proton signals were normalized by respective aromatic proton signal in the

spectra (Supplementary Fig. 1), and the DoF of gelMA was calculated as:

$$DoF[\%] = \left(1 - \frac{\text{Normalized signal (lysine methylene in gelMA)}}{\text{Normalized signal (lysine methylene in gelatin)}}\right) \times 100$$

(1)

## Microfabrication

Microfluidic devices were fabricated by a standard soft lithographic protocol[56]. Briefly, SU-8 negative photoresist (Kayaku advanced materials) was spin-coated on a silicon wafer and soft-baked on a hotplate at 95 °C. A photomask with defined device geometry was then placed atop the wafer and exposed under a collimated UV light (URE-2000/35 L, Institute of Optics and Electronics, Chinese Academy of Sciences) to cure the depicted channel area, followed by a post baking at 95 °C and SU-8 development (SU-8 developer, Kayaku advanced materials). Polydimethylsiloxane (PDMS)-based microfluidic devices were fabricated by soft lithography. PDMS elastomers (DOWSIL™ Sylgard 184 kit) were mixed with curing agents at a 10:1 ratio (w/w) before poured onto the developed master in a Petri Dish. Uncured PDMS was then degassed and incubated at 65 °C to solidify. After that, PDMS with device channel was cut out of which the inlets and outlets were punched with 0.75 mm holes. Devices and glass slides were then plasma-treated (PDC-002-HP, Harrick Plasma) for 1 min, bonded with each other, and baked at 65 °C for 10 min. For channel hydrophobic modification, the device channels were treated with 1$H$, 1$H$, 2$H$, 2$H$-perfluorooctyltriethoxysilane (Sigma-Aldrich) for 2 min.

## Scanning electron microscopy

Two sample preparation methods were employed in this work. For imaging the surface morphology of microgels, critical point drying (CPD) was used: first, carrier oil was removed, and photo-cured samples were directly resuspended in 50% ethanol. Microgels were gradually dehydrated consecutively by 75% and 100% ethanol for two days in total before CPD. Microgels were transferred into microporous specimen capsules (78 μm, Agar Scientific) that were placed in a specimen boat filled with 100% ethanol. The specimen boat was inserted into a critical point dryer (E3100, Quorum Technologies) and flushed a minimum of four times with liquid CO$_2$. Following flushing, the sample was heated to 37 °C at 80 bar pressure to dry. To image the surface morphology of the PAM scaffolds, the sample was lyophilized: briefly, the PAM scaffold was snap-frozen in liquid nitrogen and directly dehydrated in a lyophilizer overnight. For SEM imaging, samples were mounted on aluminum SEM stubs using conductive carbon sticky pads and coated with 15 nm iridium using a K575X sputter coater (Quorum Technologies). A FEI Verios 460 scanning electron microscope was utilized for imaging.

## Rheology

Rheological characterization of the jammed microgel ink and PAM scaffolds was performed using parallel plates geometry (20 mm diameter, 1 mm gap) mounted on strain-controlled rheometers (Thermo Scientific HAAKE MARS 40/60 Rheometer). Jammed microgels were prepared in the same way and molded to an identical geometry to the plates. PAM scaffolds were formed by 300 s blue light radiation. For the reversed crosslinking strategy, the jammed microgels resuspended with enzymes were allowed to anneal for 30 min in a sealed Petr Dish before measurements. Shear-thinning curves were obtained in strain rate-controlled measurements with 1% strain and shear rates from 0.01 to 10 s$^{-1}$. The recovery behavior of the ink was carried out by a step-strain sweep where low strain (1%) and high strain (90%) were cycled every 100 s with a frequency of 1 Hz. Strain sweep tests were conducted with a frequency of 1 Hz and strain from 0 to 1000%. Moduli were calculated from the linear viscoelastic region of the strain sweep diagrams. Strain sweep tests before and after annealing were not paired. Rheological characterization of the 1% CMC polymer solution was performed using the same machine with a different plate geometry (35 mm diameter, 1 mm gap). Shear-thinning curves were obtained in strain rate-controlled measurements with 1% strain and shear rates from 0.01 to 100 s$^{-1}$. Frequency sweep was conducted with 1% strain and frequency from 0 to 10 Hz. All measurements were performed at room temperature.

## Microfluidic generation of core-shell droplets

Gelatin and gelMA were dissolved in PBS to a final concentration of 15%, respectively, and kept at 37 °C to prevent thermal-induced gelation. Stock solution of 1% carboxymethylcellulose (CMC, Shanghai Aladdin Biochemical Technology) spiked with 10 U/ml transglutaminases (200 U/g, Shanghai Yuanye Biotechnology) was kept at 4 °C. For biological experiments, all polymers were dissolved in their corresponding culture medium. Next, gelatin and gelMA solutions were mixed by the ratio of 1:2 (v/v) to give a polymer blend containing 5% gelatin and 10% gelMA (referred to as the hydrogel solution), which was then spiked with 0.5% LAP photoinitiators (Shanghai Bide Pharmatech). Regardless of the involvement of cells, all polymer solutions were syringe-filtered (0.22 μm, Millex Syringe Filters) before microfluidic experiments. All solutions were loaded into 1 mL sterile plastic syringes. The hydrogel solution was placed at 37 °C and CMC 4 °C.

Core-shell droplets were generated in a flow-focusing microfluidic chip that was characterized with a channel height of 100 μm and a junction dimension of 150 × 150 μm (Supplementary Fig. 2a). Hydrogel solution and CMC were displaced into the device by syringe pumps (TYD01-02, Lead Fluid) via the side channels and the middle channel, respectively. At the device junction, CMC was engulfed by the hydrogel solution, and together they were sheared by the Novec 7500 fluorocarbon (3 M) containing 0.1% (v/v) Pico-surf surfactants (Sphere Fluidics) acting as the continuous phase. Flow rates were controlled by syringe pumps at 40, 8, 2 μL/min for oil, hydrogel solution, and CMC, respectively. As the relatively high concentration of the hydrogel solution readily caused thermal-induced gelation, a hot water bag was placed atop of the syringes and the tubing was wrapped by a custom-built heating pad.

## Fabrication of dual-network PAM scaffolds via extrusion printing

Formation of the first covalent network was catalyzed by calcium-independent transglutaminases, where the lysine and glutamine were crosslinked by a new isopeptide bond. To fully crosslink the droplets, they were incubated at room temperature overnight. Cured droplets were demulsified by adding 20% (v/v) 1$H$, 1$H$, 2$H$, 2$H$-perfluoro-1-octanol (PFO, Shanghai Bide Pharmatech) in Novec 7500 to the microgel suspension. Microgels rapidly agglomerated which facilitated the removal of most oil phase, after which they were centrifuged at 2000 rpm (~280 × $g$) for 10 min, and the residual oil at the tube bottom was aspirated. Microgels were then semi-soaked, i.e., microgels generated from 1 h of microfluidic running (600 μL dispersed phase) were directly resuspended with 150 μL PBS containing 0.5% LAP in centrifugal tubes via vortex to yield a satisfactory printability without any loss of microgels. To extrusion print microscopically heterogenous scaffolds, microgels of varying compositions, for example, functionalized with different fluorophores or containing different microbes, were mixed thoroughly together before demulsification.

To extrusion print microgels using a commercial 3D printer (EFL-BP-6601, Yongqinquan Intelligent Equipment Co., Ltd., Suzhou, China), microgel aggregates were loaded into a 5 mL printing cartridge and manually degassed. The jammed microgel ink was deposited pneumatically with a printing pressure of ~50 kPa and a 18 G printing nozzle (inner diameter: 840 μm, untapered, Nordson EFD). The writing speed was manipulated by a robotic arm and adjusted ad hoc to be in tune with the extrusion speed. Printing shapes were predefined in the user interface provided by the manufacturer of the 3D printer. Scaffolds were printed onto a clean glass slide and irradiated for 300 s by a 405 nm wavelength blue light (light intensity 25 mW/cm$^2$, Yongqinquan Intelligent Equipment Co., Ltd., Suzhou, China), resulting in interconnected and covalently stabilized microgel scaffolds. For scaffolds with the reversed crosslinking strategy, droplets were first cured by the blue light radiation for 300 s, demulsified by PFO, and resuspended with an equivalent amount of PBS spiked with 10 U/g

transglutaminases, before extruded and patterned into scaffolds and allowed for inter-microgel annealing for 30 min in a sealed Petri Dish. Fluorescent microgels were generated from a mixture of gelatin with fluorophore-labeled gelMA (green: ELF-GM-GF-60, red: ELF-GM-RF-60; Yongqinquan Intelligent Equipment Co., Ltd., Suzhou, China).

## Degradation kinetics of PAM scaffolds

Droplets of differing sizes were generated first (Supplementary Fig. 4a), where the overall flow rate of the dispersed phase was kept constant, 10 μL/min, while that of the continuous phase was varied. Droplets were collected in batches every 30 min and then incubated at room temperature overnight, after which they were demulsified by PFO and resuspended with 75 μL LAP-containing PBS. Then microgels were centrifuged at 10,000 rpm (-6900 × $g$) for 5 min and made to sediment evenly at the tube bottom. The second network was formed by blue light radiation for 300 s. To digest the PAM scaffolds, 100 μL trypsin (Beyotime Biotechnology) was layered at the top of the scaffold and incubated at 37 °C. Every 15 min the remaining scaffold was centrifuged at 10000 rpm (-6900 × $g$) for 1 min and the supernatant was aspirated, followed by an addition of 100 μL fresh trypsin and incubation. For bulk hydrogel, 240 μL hydrogel solution containing 0.5% LAP and 60 μL 1% CMC spiked with 10 U/mL transglutaminases were vigorously mixed and incubated at room temperature overnight. Before proteolytic digestion, the bulk hydrogel was also subjected to 300 s blue light radiation. For non-annealed microgels, medium-sized microgels were generated and incubated in the same fashion; before demulsification, the microgel suspension was irradiated by the blue light for 300 s and therefore the microgels were not interconnected to form a scaffold. Hydrogel weights were calculated by subtracting the weight of empty tubes and normalized by the starting weight of hydrogels:

$$\text{Normalized hydrogel weight [\%]} = \left( \frac{w_{\text{tube weight}} - w_{\text{empty tube weight}}}{w_{\text{tube weight before digestion}} - w_{\text{empty tube weight}}} \right) \times 100 \tag{2}$$

## Microbial cell encapsulation and cultivation in core-shell microgels

**Strain and medium.** The *Escherichia coli* DH5α used in this work carried a quorum-sensing plasmid (pTetR-LasR-pLuxR-eGFP) that senses the presence of signal molecule of *N*-(3-Oxodecanoyl)-L-homoserine lactone (3-Oxo-C12-HSL) and reports via the expression of eGFP, which was previously used in another work of ours[45]. Luria-Bertani (LB) medium (1 L): 10 g tryptone, 10 g NaCl, and 5 g yeast extract. The pH of the solution was adjusted to 7.0 and autoclaved before use. *E. coli* were subcultured in the LB medium before encapsulation experiments.

**Cell encapsulation and cell culture.** All polymers were dissolved by LB media. *E. coli* cells transfected with eGFP were suspended into the 1% CMC solution to a final cell concentration of OD = 0.1 for cell culture (Fig. 3). Droplets were generated and were cured immediately by blue light radiation for 300 s. Then, cell-laden microgels were demulsified and resuspended by an excessive amount of LB media spiked with 10⁻⁶ mol/L 3-Oxo-C12-HSL (Sigma-Aldrich) before incubation at 37 °C. *E. coli.* culture in non-core-shell microgels was conducted exactly in the same fashion except that the cell-containing phase was devoid of CMC. 50 μL microgels suspensions were aliquoted at 0, 12, and 24 h and imaged under an inverted fluorescence microscope (Olympus IX73).

**Quantification of cell leakage.** The amount of cells that leaked from the microgels to their medium was determined by plating experiments. After 24 hours of cell culture, 100 μL media were aliquoted and diluted by 10⁵, 10⁶, and 10⁷-fold, respectively, and then plated on LB agar plates. The CFU was calculated after 24 hours inculcation at 37 °C

by courting formed colonies on the plates. Experiments were performed in triplicate.

## Mammalian cell encapsulation and cultivation in core-shell microgels

**Cell lines and medium.** A549 and HEK 293 T were from the Narita group at University of Cambridge which were obtained from the American Type Culture Collection. Cells were cultured Dulbecco's Modified Eagle Medium (DMEM, Gibco™ 31053044), supplemented with 10% (v/v) fetal bovine serum (FBS, Gibco™ A3160801), 2 mM l-glutamine (Gibco™ 25030024), 1 mM sodium pyruvate (Gibco™ 11360070), and 0.5% (v/v) penicillin-streptomycin (Gibco™ 15070063).

**Cell encapsulation and cell culture.** All polymers were dissolved by the cell culture medium. HEK 293 T cells (10 million/mL) were suspended into the 1% CMC solution before encapsulation. For verification of the Poisson process (Supplementary Fig. 9), A549 cells were also encapsulated as a comparison. Droplets were generated by the aforementioned protocol. Cell-laden droplets were cured upon collection by 300 s blue light radiation, demulsified, and resuspended in DMEM, followed by incubation at 37 °C and 5% CO₂.

**Poisson encapsulation.** Cell-laden microgels were aliquoted and imaged after cell encapsulation. The number of cells in each microgels were manually counted and the frequency was plotted. To fit the Poisson stochasticity model, the average number of cells inside microgels was calculated and therefore the theoretical Poisson model could be plotted:

$$P(X = k) = \frac{\lambda^k e^{-k}}{k!} \tag{3}$$

In which $k$ represents the observed number of cells encapsulated in the core-shell microgels and $\lambda$ the average number of cells encapsulated in microgels.

**Live/Dead staining.** Cell viability of HEK 293 T was evaluated using Live/Dead staining kit (Invitrogen). At day 0, 3, and 6, 30 μL microgels suspension were aliquoted and incubated at 37 °C for 30 min in medium containing 4 μM calcein acetoxymethyl (calcein AM) and 2 μM ethidium homodimer-1 (EthD-1), before imaged by a Leica DMI6000B epifluorescence microscope.

**Characterization of spheroid growth.** Spheroid growth was quantified by their area and circularity all by the FIJI software. Specifically, fluorescence images of two channels, Live and Dead, were merged and thresholded into binary images, after which the area ($A$) and the perimeter ($P$) of spheroids could be directly measured by FIJI. The circularity of spheroids was calculated as:

$$\text{Circularity} = \frac{4\pi A}{P^2} \tag{4}$$

**Visualization of cytoskeleton.** 500 μL microgel suspensions were aliquoted and centrifuged at 3500 rpm (-850×$g$) for 5 min and the supernatant was aspirated. Microgels were then incubated at room temperature for 30 min with 100 μL 4% paraformaldehyde and intensively washed with PBS after incubation. Next, microgels were resuspended in 100 μL PBS with 0.1% Triton X-100 (Fisher Bioreagents BP151-500, Thermo Fisher). 4′, 6-diamidino-2-phenylindole (DAPI, Sigma-Aldrich) and Alexa Fluor™ 647 phalloidin (Invitrogen) stock solutions were added and diluted by the factor of 1000 and 40, respectively, followed by an incubation of 30 min at room temperature. Finally, microgels were washed at least twice and resuspended in PBS. Visualization of monolayer-based culture followed the same protocol. The spatial organization of nuclei and f-actin were imaged using a Leica

TCS SP8 confocal microscope. Z-channels were collected at every 10 μm for the construction of 3D structure of spheroids. The imaging of monolayered cells was conducted by a fluorescence microscope (Olympus IX73).

## Yeast-laden core-shell scaffolds for ethanol fermentation

**Strain and medium.** Yeast *Saccharomyces cerevisiae* was purchased from Fleischmann's Rapid Rise and activated in the YPD medium for 12 hours before cell encapsulation experiments. YPD medium (1 L): yeast extract 10 g, peptone 20 g, and dextrose 20 g. Culture media were autoclaved before use.

**Cell encapsulation, bioprinting, and fermentation.** All polymer solutions were dissolved by YPD media. Cell-laden core-shell droplets were generated with a yeast OD = 0.8 and incubated at room temperature overnight to be cured. Two-layered $3 \times 3$ lattice scaffolds (dimension: roughly $20 \times 20 \times 2$ mm) were hand-extruded in a biological safety cabinet and irradiated by blue light for 300 s before submersion in 1.5 mL YPD media for ethanol fermentation. The media were bubbled with nitrogen gas for 5 min to remove oxygen and the glass vials were sealed during fermentation. To generate microgels with varied sizes, two microfluidic devices and different flowrates were employed. Core-shell microgels A and B were generated with a flow-focusing device with a junction dimension of $150 \times 150$ μm and a channel height of 100 μm. Core-shell microgels C were generated with a device of the same design but with a junction dimension of $100 \times 100$ μm and a channel height of 75 μm. The volume ratio between the shell and the core phase was calculated as:

$$V_{shell}/V_{core} = \frac{R_1^3 - R_2^3}{R_2^3} \qquad (5)$$

Where $R_1$ is the diameter of the core-shell microgel and $R_2$ the core, measured by the FIJI software.

**Analytical method.** 50 μL media were aliquoted every 2 hour over 12 hours and centrifuged at 10,000 rpm ($\sim 6900 \times g$) for 5 min. The supernatant was filtered (0.22 μm) and analyzed by gas chromatography (GC, Agilent gc 6890n) using a DB-1701 column (Agilent, $30 \text{ m} \times 0.25 \text{ mm}$). Ethanol concentration was derived from an ethanol standard curve.

## Microalgae-bacteria consortium-laden core-shell scaffolds for bioremediation

**Strains and media.** Microalga *Chlorella vulgaris* was purchased from Shanghai Guangyu Biotechnology Co., Ltd and directly subcultured. Bacterium *Bacillus subtilis* was a gift from Prof. Su Chen's lab at Nanjing Tech University[3]. Before encapsulation experiments, *Bacillus subtilis* were cultured in the LB medium at 37 °C and *Chlorella vulgaris* were cultured in the BG11 medium (Qingdao Haibo Biotechnology) and incubated at 25 °C with a light intensity of ~10000 lux and cycles of light/dark conditions every 12 h (MQT-60G, Shanghai Minquan Instrument). Media recipes for bioremediation: amoxicillin bioremediation: 300 mg/L amoxicillin (Shanghai Yuanye Biotechnology) in the BG11 medium. Methyl orange bioremediation[3]: 100 mg/L methyl orange (Shanghai Aladdin Biochemical Technology), 22.16 mg/L $NH_4Cl$, 6.57 mg/L $KNO_3$, 1.08 mg/L $NaNO2$, 5.09 mg/mL $KH_2PO_4$. Amoxicillin/methyl orange was dissolved into the rest of the respective media after autoclaving, and the complete media were syringe-filtered (0.22 μm) before microfluidic experiments.

**Microbial cell encapsulation, bioprinting, and biodegradation.** All polymers were all dissolved by corresponding media without amoxicillin/methyl orange. Cell-laden core-shell droplets were generated with OD = 0.4 for both microbes, respectively, and incubated at room temperature overnight to be cured. Three-layered $3 \times 3$ lattice scaffolds (dimension: roughly $20 \times 20 \times 3$ mm) were hand-extruded in a biological safety cabinet and irradiated by blue light for 300 s before submersion in 6 mL media for bioremediation (containing amoxicillin/methyl orange for bioremediation, BG11 medium for scaffold biodegradation experiment). To print heterogeneous scaffolds, core-shell microgels encapsulating monoculture of bacteria or microalgae were mixed evenly before demulsification and printed together into scaffolds. For homogeneous scaffolds, cell suspensions were mixed at 1:1 (v/v) before microfluidic encapsulation. Scaffolds were incubated at 25 °C with a light intensity of ~10,000 lux and cycles of light/dark conditions every 12 h for bioremediation. The disintegration of scaffolds was monitored every 24 hours.

**Analytical methods.** 150 μL media were aliquoted at 6, 12, 18, and 24 h after incubation and centrifuged at 10,000 rpm ($\sim 6900 \times g$) for 5 min. The supernatant was syringe-filtered before analysis. Concentration of methyl orange was measured by a UV-Vis spectrophotometer (UV-3600, Shimadzu) and compared to a standard curve in corresponding media. Concentration of amoxicillin was measured by High Performance Liquid Chromatography (HPLC, LC-20AD, Shimadzu) equipped with a C18 column (Waters SunFire C18 column, 5 μm, $4.6 \times 250$ mm) and compared to a standard curve of amoxicillin in corresponding media. The concentration of proteases in the heterogeneous media were measured using a commercially available kit (D799673-0050, Sangon Biotech) and following the manufacturer's instruction.

## Fungi-bacteria consortium-laden core-shell scaffolds for 2-PE fermentation

**Strains.** All strains were obtained from Professor Wenming Zhang at Nanjing Tech University[60]. Strains: Fungus *M. guilliermondii* MG57: *mgpdc-mgadh-scaro10-scgap-scaro80-mggdh*. Overexpressed genes *mgpdc*, *mgadh*, and *mggdh* were from *M. guilliermondii*[57], and *scaro10*, *scgap*, and *scaro80* were from *Saccharomyces cerevisiae*[58,59]. Bacterium *Escherichia coli* YLC20: *Escherichia coli* W1485 containing plasmids for overexpressing gene *aroF* and *pheA* and for CRISPR/Cas9 knock-out of the gene *tyrA*.

**Medium.** 100× salt solution (in 1 L): 100 g NaCl, 50 g $MgCl_2 \cdot 6H_2O$, 20 g $KH_2PO_4$, 30 g $NH_4Cl$, 30 g KCl, and 1.5 $CaCl_2 \cdot 2H_2O$. Solution was autoclaved before use. Trace Element Solution (TES, in 1 L): 2 g $Al_2(SO_4)_3 \cdot 18H_2O$, 0.75 g $CoSO_4 \cdot 7H_2O$, 2.5 g $CuSO_4 \cdot 5H_2O$, 0.5 g $H_3BO_3$, 24 g $MnSO_4 \cdot 7H_2O$, 2.5 g $NiSO_4 \cdot 6H_2O$, 15 g $ZnSO_4 \cdot 7H_2O$, and 3 g $Na_2MoO_4 \cdot 2H_2O$. Synthetic medium (in 1 L): 3 g $MgSO_4$, 3 g $KH_2PO_4$, 1 g NaCl, 5 g $(NH_4)_2SO_4$, 0.015 $CaCl_2 \cdot 2H_2O$, 0.1125 $FeSO4 \cdot 7H_2O$, 1 g trisodium citrate, 10 g yeast extract, 0.3 g l-tyrosine, 0.5 g yeast nitrogen base, and 2.3 g N-[Tris(hydroxymethyl)methyl]-2-aminoethanesulfonic acid. The solution was autoclaved and then added with 45 g glucose autoclaved separately and 0.22 μm syringe-filtered 0.075 g vitamin B1 and 0.04 g kanamycin sulfate. After that, the autoclaved 100× salt solution was added into the mixture by a portion of 10 mL per 1 L medium, along with the autoclaved TES 1.5 mL/L and the volume of the mixture was adjusted to 1 L to give the complete synthetic medium.

**Microbial cell encapsulation, bioprinting, and 2-PE fermentation.** All polymers were all dissolved in the synthetic media. Cell-laden core-shell droplets were generated with OD = 0.5 for both microbes, respectively, and incubated at room temperature overnight to be cured. Five-layered $3 \times 3$ lattice scaffolds (dimension: $\sim 0 \times 20 \times 5$ mm) were hand-extruded in a biological safety cabinet and irradiated by blue light for 300 s before submersion in 8 mL media for 2-PE fermentation. For liquid culture-based 2-PE fermentation, to ensure an equivalent cell loading, the same amount of 1% CMC containing both microbes (OD = 0.5 each before mixing evenly) were aliquoted and incubated alongside droplet groups at room temperature overnight

and inoculated into 8 mL synthetic media at the same time when scaffolds were submerged. Homogeneous and heterogeneous scaffolds were fabricated the same way as the previous. All scaffolds were incubated at 37 °C for the first 24 hours and moved to 30 °C from day 2 onward. Scaffolds after a batch fermentation, e.g., the first 3 days, were carefully taken out and washed three times by fresh media, before immersed in fresh 8 mL media for reuse, which was repeated every three days for three rounds of reuse in total. 2-PE concentration was measured after each batch. Both heterogeneous and homogenous scaffolds were reused.

**Analytical method.** The kinetic curve of heterogeneous scaffolds was obtained from 6 days of monitoring 2-PE concentration at 24-hour intervals. The comparison among heterogeneous scaffolds, homogeneous scaffolds, and liquid culture was made at the end of day 3. 150 μL media were aliquoted and centrifuged at 10,000 rpm (~6900× g) for 5 min. Supernatants were syringed-filtered (0.22 μm) before analysis. The concentration of 2-PE was measured by GC (Agilent gc 6890n) equipped with a DB-1701 column (Agilent, 30 m × 0.25 mm) and compared with a standard curve.

### Statistics and reproducibility

All data handling and statistical analysis were performed by Microsoft Excel 2019 and plotted using Python (v. 3.8.12) scripts using NumPy (v.1.22.1) and Matplotlib (v.3.5.1) libraries. Data are presented as mean ± standard deviation from at least three biological replicates or otherwise stated in the manuscript. All images were processed by FIJI-ImageJ (v 2.0.0-rc-69/1.52i) and Adobe illustrator 2019 was used to design illustrations and curate figures.

Photographs, i.e., Fig. 1c, Supplementary Figs. 6a, 16a, are presentative images of printed scaffolds. Micrographs of droplets and microgels, Figs. 1b, 3a, b, 2e, 4b and Supplementary 2b, 3a, b, 7a, 9a, 9c, are representative images of randomly aliquoted samples during one of the independently repeated experiments. Micrographs of printed scaffolds, i.e., Figs. 1c, 2d, f and Supplementary Figs. 3c, 8a–c, are representative images of a local structure of scaffolds that had similar morphology. All rheological characterization was performed in triplicate and Fig. 2a, b and Supplementary Fig. 5a–e, g–i are presentative plots from similar results. The cellular spheroid data were generated from randomly aliquoted microgel samples in one experiment and Fig. 3f and Supplementary Fig. 10 are representative confocal microscopic images of spheroids with similar cellular structures. All microbial bioprocessing experiments were performed in triplicate and Supplementary Fig. 13 are one representative set of images from the experiment of scaffold degradation by the microalgae-bacteria consortium.

### Reporting summary

Further information on research design is available in the Nature Portfolio Reporting Summary linked to this article.

## Data availability

All data generated in this study are provided in the Source Data file. Source data are provided with this paper.

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

## Acknowledgements

National Key Research and Development Program of China (2021YFC2104300), the National Natural Science Foundation of China (21901117, 32111530117), and State Key Laboratory of Materials-Oriented Chemical Engineering (KL20-02) to Z.Y. The Newman Foundation, the Wellcome Trust, the European Research Council under the European Union's Seventh Framework Programme (FP7/2007-2013) through the ERC Grant PhysProt (Agreement No. 337969) to T.P.J.K. Cancer Research UK (CRUK) Cambridge Institute Core Grant (C9545/A29590) to M.N.; CRUK Early Detection Pump Priming awards (C20/A20976) to T.K. and M.N. Chinese Scholarship Council (Y.O and H.Z.).

## Author contributions

Y.O., T.P.J.K., and Z.Y. conceived and designed the experiments. Y.O. performed experiments, curated data, and wrote the manuscript. S.C., Yang Z., H.Z., and C.G. performed experiments. W.Y. and F.X. provided the information and instructions on the experiments with metabolically engineered microorganisms. M.N. provided mammalian cell lines and tissue culture facility. W.D. and Yanli Z. helped with the revision of the paper. T.K. and Z.Y. supervised the project and revised the manuscript. All authors reviewed and agreed to the final version of the paper.

## Competing interests

The authors declare no competing interests.
