## [Peer Review File · Nature Communications]

REVIEWER COMMENTS

Reviewer #1 (Remarks to the Author):

This work presented in the manuscript is very interesting and many aspects represent new ideas to the living materials field. Foremost, microgel-enabled 3D bioprinting is a fairly new research direction as well as relatively for printing microorganisms, and their work certainly demonstrates a viable and powerful combination of these two topics. Though the material is quite normal in biomaterial research, the described gelMA microgel double covalent network method is also a different approach, and I'm impressed to see it worked so well. The core-shell strategy is new to bioprinting as well. We've seen many publications of mammalian cell culture in core-shell microgels but working with bacteria is new to me. Finally, the most interesting part of this research is the micro-compartmentalization of microbial consortia in the macroscopic materials to increase the performance of the materials in bioprocessing, which is a completely novel idea in living materials. In conclusion, the authors have presented an elegant and effective bottom-up method to construct living materials, with different levels of control over the morphologies of the final materials by microfluidics and bioprinting. I think this work is innovative and can inspire other works in the emerging field of living materials and therefore deserves a publication in Nature communications. I enjoyed reading this manuscript in general, but with some questions and minor suggestions:

1. Regarding the material, I noticed that the authors mixed gelMA with gelatin. As the functionalization was not complete, there should be sufficient lysine in gelMA for enzymatic crosslinking. Why not directly using gelMA instead of this seemingly more complicated methods?
2. I noted that this work used a slightly different method for 3D printing preparation: the microgels were not fully swollen before printing. What is the rationale behind this? Also regarding the 1:4 ratio between microgels and liquid added before printing (manuscript line 107), how was this ratio obtained?
3. The authors showed that the gelMA double network method can be reversed and examined the rheology of both; but considering the covalent nature of these two gelation methods, why was there a significant difference of the storage modulus between these two settings, namely one over 4000Pa and the other less than 2000Pa.
4. In my opinion, one disadvantage of this material system is that enzymatic curing takes a long time, as the droplets were crosslinked overnight. This might bring up a concern because the microorganisms can proliferate significantly overnight and might contribute to varying cell loading before printing. Were there any measures taken to prevent the data uncertainties that might have arisen from this protocol, especially in figure 4.H the authors compared the production of 2-PE from the scaffold with the liquid culture.
5. I noticed that no specific papers were referenced for the e. coli and yeast consortia. Therefore, I suggest adding more details to the supporting materials in regard to for example the plasmid construction etc.

Reviewer #2 (Remarks to the Author):

The authors did a great job presenting their design and data. They developed a method to print protein-based core-shell microgels and used it to create functional living materials. The printed structures could host bacterial and mammalian cells in predefined

compartments. They showed that this printing method could facilitate the division of labor between cell communities and enhance their performance in bioprocessing. Although I enjoyed reading this paper, I was not fully convinced about the usability of this method in more realistic settings. As the authors stated several times, the printed structures lack mechanical robustness and could not prevent cell leakage after incubation. It is thus difficult to decouple the contribution to bioprocessing of the encapsulated cells from the cells in the liquid. Here are a few points that need to be addressed to improve the manuscript before publication.

1. Line 58: Please provide references related to these challenges.
2. Please enlarge the texts in the main and supplementary figures, so they are readable to readers.
3. Figure 1A: The functional groups should be made bigger and represented by colors with higher contrast. Also, it is unclear why the "first covalent crosslinking" figure has a purple background instead of a white one.
4. The degree Celsius signs are not displayed properly in the supplementary information document.
5. Confusingly, the authors use more asterisks to represent higher P values throughout the manuscript. Please consider the following example:
<https://www.graphpad.com/support/faq/what-is-the-meaning-of--or--or--in-reports-of-statistical-significance-from-prism-or-instat/>
6. Line 201: Is the core aqueous in all cases presented in this paper? Is there a reason why not to use hydrogels? Please consider providing more details about the properties of the core.
7. Figure S8B: This figure panel is difficult to understand without a negative control. Also, measuring GFP absorption in the media is not an ideal method to quantify cell leakage. Normally, bacterial cell leakage is measured by plating, followed by reporting CFU count and the lower limit of detection.
8. Figure S10A: Please define k in the caption.
9. Figure 4A: Please define what the triangles and circles are.
10. It seems like the printed scaffolds are relatively fragile after incubation. Could the authors please provide some simple characterization of these printed threads against physical insults such as pulling and shaking in liquid (before and after incubation)?
11. Line 263-273: Could the authors provide more evidence to prove that secreted proteases caused the disintegration? The link between scaffold disintegration and the bioactivity of the cells seems rather speculative and weak. There are many ways to directly characterize the metabolic state of cells and the presence of secreted proteases.
12. Line 299-319 and Figure S15: With the amount of cell leakage shown in these photos, there were likely more cells in the liquid media than in the printed structures. Thus, the reported numbers here are unlikely to reflect the true performance of these materials in producing 2-PE.
13. The authors should discuss more specifically how the mechanical properties of these printed structures could be improved. I believe that the foundation of spatially organizing cell communities is a robust physical separation mechanism that prevents cell leakage, especially for any bioprocessing and fabrication applications. In the use scenarios proposed by the authors (incubating in growth media), even a single leakage event would cause unwanted cell growth in the surrounding media, defeating the purpose of creating heterogeneity in the first place.
14. Ref 31 and 32 are the same.

Reviewer #3 (Remarks to the Author):

The authors present in this publication a fabrication method based on microfluidics, aqueous two-phase separation (unless I have misunderstood this), and covalent cross-linking to generate micron-sized core shell hydrogel particles (with presumably a liquid interior). Cross-linking of the gelatin shell occurs via transglutimases (diffusing from the core) linking together lysine and glutamine. Core-shell microgels could be jammed (by centrifugation and rehydration) and extrusion printed (as they behaved like a shear thinning material), which could be followed by further cross-linking by the particles via UV-light, Gel-MA in the hydrogel shell and a photoinitiator. They demonstrate the growth of bacteria, microalgae, fungi, and human embryonic kidney cells, where their growth is mostly sterically confined to the interior of the particle (over shorter timescales) by the nanoporous hydrogel shell. However, over time, certain microalgae can degrade the structures completely. Combined with the 3D printing the authors show they can build living materials from this fabrication method. They utilise this method to fabricate two microbial co-culture systems, with the first comprising of *Cholera vulgaris* and *Bacillus subtilis* to degrade methyl orange and amoxicillin, and the second comprising *Escherichia coli* and *Meyerozyma guilliermondii* to generate 2-phenylethanol from glucose.

Overall, the paper has some interesting results, including a different strategy for building living materials, as compared to the literature, via microfluidics and 3D printing. However, I think this needs to be fleshed out quite a bit for a publication at Nature Communications, when compared to current living material papers published in this journal. The crux of the paper is based on bioprinting, yet there does not seem to be a demonstration of the advantage to print these structures. In general, the data does support the conclusions, but often I had to go to the supplementary methods to understand the system. Moving parts of the methods into the main text would be useful in some parts and I will highlight this later. Importantly, the ecological terms used in this paper are sometimes not accurate of the ecological literature and need to be corrected to not mislead the reader.

Comment 1: When generating the core-shell hydrogel structures via microfluidics the authors briefly mention in line 66-67 that this an aqueous two-phase system. I presume this is between the gelatin and the and the carboxymethylcellulose. Can the authors clarify this, and if so demonstrate this behaves as an aqueous two-phase system as compared to liquid-liquid phase separation papers in the literature? If I have understood this incorrectly, can the authors add more detail why the two aqueous phases do not mix before the hydrogelation of the shell.

Comment 2: In Fig. 1, the schematic looks like crosslinking is occurring everywhere which is slightly confusing as from the text it should only be happening in the shell. This is obviously not the authors intention.

Comment 3: Can the authors comment more on the environment of the core as presumably this is a viscous liquid? It will be important for growth of the microorganisms and cells as it will determine whether they are sedimented at the bottom and growing, or indeed growing in 3D. A confocal z-stack would be useful to show this compared to the x,y images shown in Fig. 3 for *E. coli* and HEK cells.

Comment 4. Minor point, line 200, this is not bacterial community but a bacterial population as it is a single species (*E. coli*). Minor points for the word symbiosis, which is used in line 257 and lines 337-338. Symbiosis is often defined as a long-term relationship between two organisms that often needs a host. My advice would be to remove this term as it unlikely to apply to these microbial systems.

Comment 5. For the relationship between *C. vulagris* and *B. subtilis*, it is not clear how they degrade amoxicillin or methyl orange, or why you need both of them to do this. Can this be

clarified in the text. Moreover, reference 37 points to this relationship between *C. vulgaris* and *B. subtilis* for bioremediation of organic contaminants – I could not find this in the reference (maybe I missed this). Can the authors provide other experimental evidence for this or another reference that explicitly shows this? In line 257, the authors write that the relationship between these two microbes are a naturally occurring symbiosis. Can the authors provide either a strong reference for this or experimental evidence for their interactions via pairwise cultures (see reference: Mitri S, Foster KR (2013). *Annual Review of Genetics*,47:247–73, figure 5 for an explanation on how to do this).

Comment 6. For the experiments related to Fig. 4C important controls are missing such as homogenous PAM scaffolds (i.e. *C. vulgaris* and *B. subtilis* co-encapsulated), and scaffolds just containing either *C. vulgaris* and *B. subtilis*, and liquid culture (i.e., what has been similarly done for Fig. 4h). It is important to show that heterogenous PAM scaffolds, i.e., segregating out *C. vulgaris* and *B. subtilis* is beneficial for bioremediation. The authors allude to this later in the paragraph by a degradation assay, however this does not directly measure bioremediation. This is important to show, as if this cannot be shown, then the fabricated system is not useful for this assay. One minor point is that it is not clear what is the morphology and dimensions of 3 x 3 lattice shaped PAM scaffold. Can the authors include a diagram or photograph of this. Was this 3 x 3 lattice used in the bioremediation assay as well as the degradation assay?

Comment 7. Line 299 the authors claim the *Escherichia coli* and *Meyerozyma guilliermondii* form a competitive relationship. From Fig. 4h on co-encapsulation, the production of 2-PE is lower than when segregating the species, however, cell numbers are not measured, so competition between the species cannot be inferred. Please confirm this from the literature or demonstrate via a liquid pairwise competition assay mentioned in comment 5.

Comment 8. Line 304, the authors claim the endpoint production of 2-PE at 14.23 ± 1.12 mg mL⁻¹ is among the highest in the literature. Can references and comparison values be provided.

Comment 9. The two microbial assays in Fig. 4 do not necessarily directly demonstrate the benefit of the core-shell morphologies. This may be inferred that the microorganisms will grow poorly (and therefore may have difficult performing bioremediation of 2-PE synthesis) from the *E. coli* results in Fig. 2B, however a direct comparison would be useful i.e. showing how these microbes encapsulated in homogenous microgels, printed, and cross-linked, perform compared to the core-shell morphologies.

Comment 10. The authors do not seem to clearly demonstrate the benefit of 3D printing for their microbial assays compared to moulding. i.e. the segregation of different microbial species is generated from randomly mixing the core-shell hydrogels rather than in the printing itself. The authors need to clearly demonstrate the benefit of 3D printing here to warrant this a useful technique. One suggestion would be to create lattices similar to what is seen in Fig. 1c and change the area of free space in-between the lattice to see if there is a difference in bioremediation or production 2-PE. Other suggestions are welcome to demonstrate the importance of 3D printing in this system.

Comment 11. Another important point of the paper is confinement of the microbes at the microscale is important for the bioactivities of the microbes (as per line 31 in the abstract). Literature definitely points to this; however, this is not explicitly demonstrated in this paper. However, it might be possible to show this. Is it possible to create larger sized-core shell particles to show a difference in bioactivities, i.e., mm-sized, or at least larger or smaller sized micron-sized particles?

Reviewer #4 (Remarks to the Author):

This work reports an approach for constructing living materials via a common extrusion bioprinting routine. In this approach, jammed core-shell microgels with bacteria or mammalian cells in the aqueous cores are interparticle annealed to give covalently stabilized functional scaffolds with controlled microporosity that enhances the mass transfer of nutrients and metabolites. In general, the topic is interesting, and the manuscript is well-written. However, it seems that the advantage of the proposed strategy in this work is not so obvious compared to those in the previous reports (e.g. Highley, C. B. et al., *Adv. Sci.* 6, (2019); Chai, N. et al., *COMPOSITES PART B-ENGINEERING*, 2021, 109100). As such, I recommend a major revision before its acceptance in the high-quality journal of *Nature Communications*. Other specific comments are listed as below.

- 1. As the authors mentioned in the manuscript, several works related to jammed microgel printing have been reported (Highley, C. B. et al., *Adv. Sci.* 6, (2019)). Even the similar method for core-shell microgel printing has been proposed (Chai, N. et al., *COMPOSITES PART B-ENGINEERING*, 2021, 109100). So, the authors should provide more evidence or data to prove the significant advantages of this manuscript compared to the previous reports.**
- 2. In regard to the characterization of the microgels, the authors should provide more data to demonstrate that the cores are liquid rather than solid. In addition, in Fig S2, it seems the compartmentalization of the microgels are not obvious. The authors should provide images with higher definition.**
- 3. In the paper, the authors claimed that the dual covalent strategy is reversible, i.e., droplets can be first cured by blue light radiation and then annealed enzymatically. However, the enzymatic reaction needs overnight incubation at room temperature. Such a slow reaction is not suitable for the fast-printing process. It seems like a limitation of the proposed method.**
- 4. As the dual-reaction process is crucial for the formation of printed scaffolds, the authors should provide more data to prove the existence of the enzymatic and photo-initiated bonds in the microgels.**
- 5. The authors have fully demonstrated the application of bacteria-laden scaffolds in bioremediate organic contaminants. What are the application scenarios for cell-laden scaffolds?**
- 6. Fig 3A, it seems that the fluorescence intensity in microgels is homogeneous at 0h, why?**
- 7. What is the exact dimension of the cellular spheroid in Fig S11? The dimensions are inferred about 100 micrometers from other figures. Generally, cellular spheroids at such dimensions seldom show necrotic cores. In addition, the dead cells randomly distributed in the cellular spheroid. Do these phenomena indicate that the presence of microgels decrease the cell viability? The authors should provide more control data to confirm the doubt.**

Bioprinting microporous functional living materials from protein-based core-shell
microgels

Point-by-point responses to reviewers

**Reviewer 1:** *“This work presented in the manuscript is very interesting and many
aspects represent new ideas to the living materials field. Foremost, microgel-enabled
3D bioprinting is a fairly new research direction as well as relatively for printing
microorganisms, and their work certainly demonstrates a viable and powerful
combination of these two topics. Though the material is quite normal in biomaterial
research, the described gelMA microgel double covalent network method is also a
different approach, and I’m impressed to see it worked so well. The core-shell
strategy is new to bioprinting as well. We’ve seen many publications of mammalian
cell culture in core-shell microgels but working with bacteria is new to me. Finally, the
most interesting part of this research is the micro-compartmentalization of microbial
consortia in the macroscopic materials to increase the performance of the materials
in bioprocessing, which is a completely novel idea in living materials. In conclusion,
the authors have presented an elegant and effective bottom-up method to construct
living materials, with different levels of control over the morphologies of the final
materials by microfluidics and bioprinting. I think this work is innovative and can
inspire other works in the emerging field of living materials and therefore deserves a
publication in Nature communications. I enjoyed reading this manuscript in general,
but with some questions and minor suggestions.”*

**Response:** We thank the reviewer for the time taken for reviewing our work and the
appreciation of our work.

**Reviewer 1:** *“1. Regarding the material, I noticed that the authors mixed gelMA with
gelatin. As the functionalization was not complete, there should be sufficient lysine in
gelMA for enzymatic crosslinking. Why not directly using gelMA instead of this
seemingly more complicated methods?”*

**Response:** Thank you for this question. Initially we attempt to use gelMA as the only
material. However, we obtained better results with the mixture of gelatin and gelMA.

It is likely that with this degree of functionalization and polymer concentration, the
gelMA material still lacks sufficient lysine residues for crosslinking and therefore we
chose to mix the gelatin and gelMA to enhance the enzymatic crosslinking step.

**Review 1:** *“2. I noted that this work used a slightly different method for 3D printing
preparation: the microgels were not fully swollen before printing. What is the
rationale behind this? Also regarding the 1:4 ratio between microgels and liquid
added before printing (manuscript line 107), how was this ratio obtained?”*

**Response:** We considered obtaining microgels fully swollen in PBS and then to
vacuum filter them before extrusion printing, which is similar to other published works
(C. B. Highley, et. al., 10.1002/adv.201801076). However, the vacuum filtration has
the potential to decrease cell viability (D. L. Alge, et al., 10.1126/sciadv.abk3087). As
stated in our manuscript, we found that the interstitial water content impacts printing,
and this ratio obtained from our optimization process represents an optimized
amount of water that would lubricate the movement of microgels during extrusion but
not enough to completely dissipate the frictions by solvents and therefore improved
printability.

**Reviewer 1:** *“3. The authors showed that the gelMA double network method can be
reversed and examined the rheology of both; but considering the covalent nature of
these two gelation methods, why was there a significant difference of the storage
modulus between these two settings, namely one over 4000Pa and the other less
than 2000Pa.”*

**Responses:** We thank the reviewer for the question. The difference in the storage
modulus arises from two reasons: first, the enzymatic annealing of photo-crosslinked
microgels took less time than the enzymatic curing of droplets (30 min versus
overnight), as in the former case, enzymes were evenly distributed in the interstitial
space among microgels while in the core-shell droplets, enzymes need to diffuse into
the shell phase to solidify the material. Second, the photo-initiated free radical
polymerization not only happens outside the microgels but also inside microgels,
which further increases the stiffness of the scaffold; however, due to the larger
molecular size of transglutaminases, in enzymatic annealing, the enzyme might not

be able to diffuse into the microgels and therefore less crosslinking density in this
scenario.

**Reviewer 1:** *“4. In my opinion, one disadvantage of this material system is that*
*enzymatic curing takes a long time, as the droplets were crosslinked overnight. This*
*might bring up a concern because the microorganisms can proliferate significantly*
*overnight and might contribute to varying cell loading before printing. Were there any*
*measures taken to prevent the data uncertainties that might have arisen from this*
*protocol, especially in figure 4.H the authors compared the production of 2-PE from*
*the scaffold with the liquid culture.”*

**Responses:** Good question. The additional incubation time might allow for
differences in cell proliferation. We have, however, taken additional measures to
ensure the precision of the data. First the cell density in cell reservoir (the syringe)
was adjusted to be the same in all cases, and we timed the droplet generation
precisely (typically 90 minutes) to make sure that we generated the same amount of
microgels and therefore cells for printing. Second we made the incubation time for
the droplet to solidify identical, and so was the incubation environment. Lastly the
scaffolds printed had a similar dimension and were weighed individually before
incubation. As for the comparison with the liquid culture, to maintain the same
conditions, microorganisms of the same amount to that immobilized in microgels
were inoculated into the CMC (the core phase solution) at the same time alongside
the cell-laden droplets and incubated together, and inoculated to the same amount of
medium the second day at the same time.

**Reviewer 1:** *“5. I noticed that no specific papers were referenced for the e. coli and*
*yeast consortia. Therefore, I suggest adding more details to the supporting materials*
*in regard to for example the plasmid construction etc.”*

**Response:** We have now added more information to the revised manuscript and we
thank the reviewer for the suggestion.

**Reviewer 2:** *“The authors did a great job presenting their design and data. They*
*developed a method to print protein-based core-shell microgels and used it to create*
*functional living materials. The printed structures could host bacterial and*
*mammalian cells in predefined compartments. They showed that this printing method*
*could facilitate the division of labor between cell communities and enhance their*
*performance in bioprocessing.”*

**Response:** We thank the reviewer for their positive comments.

**Reviewer 2:** *“Although I enjoyed reading this paper, I was not fully convinced about*
*the usability of this method in more realistic settings. As the authors stated several*
*times, the printed structures lack mechanical robustness and could not prevent cell*
*leakage after incubation.”*

**Response:** We thank the reviewer for the comments and insights. Improving the
mechanical properties of engineered living materials to circumvent cell escape is
indeed a very important aspect in designing such materials but remains challenging,
as discussed extensively in a very recent review (X. Liu, et
al., 10.1002/adma.202201326). First, hydrogel mechanical properties are inherently
associated with the manufacturability of the material (for example printability) and
affect to a great extent the fate of the cells embedded, which has long been
recognized for animal cells (B. M. Baker and C. S. Chen, 10.1242/jcs.079509) and
recently found for microorganisms as well (S. Bhusari, et
al., 10.1002/advs.202106026, and X. Liu, et al., 10.1002/adma.202201326). As a
result, striking a balance among material mechanical property, processability, and
suitability to cells has always been a trade-off and a challenge not fully addressed
yet (L. Ouyang, et al., 10.1126/sciadv.abc5529; M. Wang, et al., 10.1038/s41467-
022-31002-2; these two papers describe two different strategies of how to alter the
mechanical properties, typically to reduce the stiffness of printed cell-laden
structures after bioprinting). Therefore, the main objective of our work is to address
these challenges through a novel method of core-shell microgel printing which
decouples the material processability from cell culture microenvironment, rather than
to optimize the mechanical property of the printed scaffolds, although we agree that
the latter is also an important challenge. Please find further information to this point.

**Reviewer 2:** *“It is thus difficult to decouple the contribution to bioprocessing of the*
*encapsulated cells from the cells in the liquid. Here are a few points that need to be*
*addressed to improve the manuscript before publication.”*

**Response:** We thank the reviewer for this comment. Using existing methods, cell
leakage is currently an inevitable and frequently observed problem in bioprinting
hydrogel-based living materials. We have shown in our manuscript that our core-
shell strategy can in many cases mitigate this problem when compared with non-
core-shell microgels. Please find more information in the response.

**Reviewer 2:** *“1. Line 58: Please provide references related to these challenges.”*

**Response:** This is a very good suggestion. Following the reviewer’s comment, we
now have inserted three references related to these challenges, namely, (1) A. Daly,
et al., [10.1016/j.cell.2020.12.002](https://doi.org/10.1016/j.cell.2020.12.002), a review that has discussed the importance of
cellular microenvironments and heterogeneous structures in bioprinting; (2) Y. S.
Zhang, et al., [10.1038/s43586-021-00073-8](https://doi.org/10.1038/s43586-021-00073-8); (3) C. B. Highley, et
al., [10.1002/advs.201801076](https://doi.org/10.1002/advs.201801076), the first research paper demonstrating the extrusion
printing jammed microgels, where the authors mentioned the potential of microgels
to “permit control over cellular microenvironments through microgel design” though
they did not demonstrate it in this paper.

**Reviewer 2:** *“2. Please enlarge the texts in the main and supplementary figures, so*
*they are readable to readers.”*

**Response:** We have followed this helpful advice and made the changes accordingly
throughout the manuscript and supporting information.

**Reviewer 2:** *“3. Figure 1A: The functional groups should be made bigger and*
*represented by colors with higher contrast. Also, it is unclear why the “first covalent*
*crosslinking” figure has a purple background instead of a white one.”*

**Response:** We thank the reviewer for the suggestion. We have made the functional
groups bigger and changed the color scheme in the Fig. 1a (please also see the

attached figure below), along with other changes to make the scheme more
 readable.

 **Figure R1.** The schematic for the formation of PAM scaffolds.

**Reviewer 2:** “4. The degree Celsius signs are not displayed properly in the
 supplementary information document.”

**Response:** We have corrected the degree Celsius signs and thank the reviewer for
 pointing this out.

**Reviewer 2:** “5. Confusingly, the authors use more asterisks to represent higher *P*
 values throughout the manuscript. Please consider the following example:
 [https://www.graphpad.com/support/faq/what-is-the-meaning-of--or--or--in-reports-of-](https://www.graphpad.com/support/faq/what-is-the-meaning-of--or--or--in-reports-of-statistical-significance-from-prism-or-instat)
 [statistical-significance-from-prism-or-instat](https://www.graphpad.com/support/faq/what-is-the-meaning-of--or--or--in-reports-of-statistical-significance-from-prism-or-instat)”

**Response:** We have made this correction and thank the reviewer for pointing this
 out.

**Reviewer 2:** “6. Line 201: Is the core aqueous in all cases presented in this paper?
Is there a reason why not to use hydrogels? Please consider providing more details
about the properties of the core.”

**Response:** This is a very good question. The core phase in all cases presented in
our work is aqueous but crucially consists of a highly viscous polymer solution. To
enhance clarity we have changed the term throughout the manuscript. As for the
rationale behind the use of viscous solution instead of hydrogels, our idea is to use
the phase behavior between gelatin/gelMA and CMC (please see comment 1
reviewer #3) to induce the core-shell structure amid microfluidic confinement of cells
inside the core, which decouples the cell culture environment from the inherent
mechanical properties of the shell hydrogel material and gives rise to more
controllability over the cellular microenvironments. Further, recent work shows that
high stiffness of hydrogel impacts the metabolism of bacteria embedded within (S.
Bhusari, et al., 10.1002/adv.202106026) and we have also found that the core-shell
strategy improves the anaerobic fermentation of yeast in comparison with non-core-
shell microgels (please see comment 9, reviewer #3) as well as mitigates cell
leakage (please see comment 7).

To directly probe the liquid nature rather than a crosslinked hydrogel network, we
added some extra SEM characterization of the core-shell microgels (see attached
figure R2A), which demonstrates that after the dehydration of microgels, they display
a solid shell with a hollow core. Regarding the properties of the core phase, we
further conducted rheological characterization of 1% carboxymethylcellulose (CMC)
solution (see attached figure R2B-D), which shows that under low testing frequency,
the loss modulus of the CMC is higher than its storage modulus, suggestive of a
more viscous behavior instead of elasticity when the material is static. Moreover,
both the viscosity-shear rates and shear stress-shear rates plots (figure R2C-D)
indicate that the material is shear-thinning, in agreement with previously reported
rheological results (A. Benslimane, et al., 10.1016/j.clay.2016.08.026).

Figure R2. SEM images of core-shell microgels and rheology of CMC (1%).

Reviewer 2: “7. Figure S8B: This figure panel is difficult to understand without a
 221 negative control. Also, measuring GFP absorption in the media is not an ideal
 method to quantify cell leakage. Normally, bacterial cell leakage is measured by
 plating, followed by reporting CFU count and the lower limit of detection.”

Response: We have followed the helpful advice and performed a plating experiment
 with core-shell microgels and non-core-shell microgels (please see attached results),
 and we again found that the core-shell strategy would mitigate the cell escaping
 tissue. As a result we have replaced the GFP adsorption results accordingly and
 thank the referee for this suggestion.

Figure R3. Cell leakage and plating experiments.

**Reviewer 2:** "8. Figure S10A: Please define k in the caption."

**Response:** We have done this and thank the reviewer for this suggestion.

**Reviewer 2:** "9. Figure 4A: Please define what the triangles and circles are."

**Response:** We have added more information in the figure caption as suggested by
the reviewer.

**Reviewer 2:** *“10. It seems like the printed scaffolds are relatively fragile after*
*incubation. Could the authors please provide some simple characterization of these*
*printed threads against physical insults such as pulling and shaking in liquid (before*
*and after incubation)?”*

**Response:** We have followed the reviewer’s helpful advice and produced a video to
show that the scaffold was stable after three days incubation. Please check the
additional video file.

**Reviewer 2:** *“11. Line 263-273: Could the authors provide more evidence to prove*
*that secreted proteases caused the disintegration? The link between scaffold*
*disintegration and the bioactivity of the cells seems rather speculative and weak.*
*There are many ways to directly characterize the metabolic state of cells and the*
*presence of secreted proteases.”*

**Response:** We thank the reviewer for the question and for the helpful suggestions
that we have followed. First, we have conducted a new experiment to quantitate the
concentration of proteases using a commercially available protease detection kit.
Results show that proteases were present in the medium with the concentration
increasing over time. Second, we have also added a new experiment using the
spatial segregation strategy to bioremediate methyl orange, which demonstrates that
the heterogeneous scaffolds can remediate more methyl orange compared to other
scaffolds in 24 hours. Please see attached results.

Figure R4. Detection of proteinases and methyl orange bioremediation.

**Reviewer 2:** “12. Line 299-319 and Figure S15: With the amount of cell leakage
 shown in these photos, there were likely more cells in the liquid media than in the
 printed structures. Thus, the reported numbers here are unlikely to reflect the true
 performance of these materials in producing 2-PE.”

**Response:** We thank the reviewer for the question. To the best of our knowledge,
 there is currently no effective method, in the field of extrusion bioprinting
 microorganism-laden hydrogels for bioprocessing/biomanufacturing purpose, to
 completely prevent cell leakage to the media. There are a number of previous works
 that have reported this problem, for example, T. Butelman et
 al., 10.1021/acsabm.1c00754; T. G. Johnston, et al., 10.1038/s41467-020-14371-4;
 H. Priks, et al., 10.1021/acsabm.0c00335; A. Saha, et al., 10.1021/acsami.8b02719;
 280 M. Schaffner, et al., 10.1126/sciadv.aao6804.

As such, the focus of our work is to introduce a new way of fabricating functional
 living materials using a new way of spatially segregating cells to improve
 bioprocessing. In order to showcase the advantage of our method, we compared the
 production of heterogeneous scaffolds with homogeneous scaffolds, and also with
 traditional liquid culture, and the results show that the 2-PE production was
 significant higher using our method, suggestive of an improvement resulting from the

spatial arrangement of cells. We believe that this new technique can be generalized
and mirrored to other material systems, preferably synthetic materials such as
PEG/Dextran (S. Ma et al., 10.1002/sml.201102715) that have a wider range of
mechanical tunability and potentially insusceptibility to biodegradation.

Following the reviewer's helpful advice, we have changed the description from
"heterogeneous scaffolds produced more 2-PE than..." to "a higher amount of 2-PE
was detected from the fermentation broth containing heterogeneous scaffolds" or
other similar descriptions. Lastly, we are now also working on strategies to prevent
microbes immobilized in microgels from escaping, for example by surface coating (S.
Zhao, et al., 10.1016/j.snb.2021.129648), and we surely envisage the emergence of
a more mechanically robust material system that can be used for bioprinting.

**Reviewer 2:** *"13. The authors should discuss more specifically how the mechanical*
*properties of these printed structures could be improved. I believe that the foundation*
*of spatially organizing cell communities is a robust physical separation mechanism*
*that prevents cell leakage, especially for any bioprocessing and fabrication*
*applications. In the use scenarios proposed by the authors (incubating in growth*
*media), even a single leakage event would cause unwanted cell growth in the*
*surrounding media, defeating the purpose of creating heterogeneity in the first*
*place."*

**Response:** We thank the reviewer for this insight and we have added some thoughts
about possible developments that concern mechanical properties but also further
factors, as improving mechanical properties of the hydrogel alone is not in all cases
likely to be the most effective solution to cell leakage. To expand on this idea, we
now have added a new result in our revised manuscript to show that in the non-core-
shell microgel scaffold, where the yeast cells were embedded in a chemically
crosslinked hydrogel network, the ethanol fermentation process was severely
hampered (reviewer #3, comment 9), suggesting that the yeast cells grew poorly in
such environment. As such, there is a balance between mechanical stability and
functionality. Our core-shell platform allows us to improve both factors relative to
previous approaches.

As such, the main aim of this study is proposing a new idea for such applications.
Moreover, despite the residual cell leakage to the medium, we noted a significantly
higher concentration of 2-PE by heterogeneous scaffolds than by both homogenous
scaffolds and the liquid culture, which demonstrates the potential of our method.
Further, the cell leakage does not happen at the initial stages of cell culture but after
a certain period of incubation and the core-shell strategy can mitigate this process in
comparison with homogeneous microgels (comment 7), which in our opinion is
another advantage of our method. Following the reviewer's helpful advice, we have
added more discussions to the revised manuscript.

**Reviewer 2:** *"14. Ref 31 and 32 are the same."*

**Response:** We have corrected it and thank the reviewer for noticing this.

**Reviewer 3:** *“The authors present in this publication a fabrication method based on*
*microfluidics, aqueous two-phase separation (unless I have misunderstood this), and*
*covalent cross-linking to generate micron-sized core shell hydrogel particles (with*
*presumably a liquid interior). Cross-linking of the gelatin shell occurs via*
*transglutimases (diffusing from the core) linking together lysine and glutamine. Core-*
*shell microgels could be jammed (by centrifugation and rehydration) and extrusion*
*printed (as they behaved like a shear thinning material), which could be followed by*
*further cross-linking by the particles via UV-light, Gel-MA in the hydrogel shell and a*
*photoinitiator. They demonstrate the growth of bacteria, microalgae, fungi, and*
*human embryonic kidney cells, where their growth is mostly sterically confined to the*
*interior of the particle (over shorter timescales) by the nanoporous hydrogel shell.*
*However, over time, certain microalgae can degrade the structures completely.*
*Combined with the 3D printing the authors show they can build living materials from*
*this fabrication method. They utilise this method to fabricate two microbial co-culture*
*systems, with the first comprising of Cholera vulgaris and Bacillus subtilis to degrade*
*methyl orange and amoxicillin, and the second comprising Escherichia coli and*
*Meyerozyma guilliermondii to generate 2-phenylethanol from glucose.*
*Overall, the paper has some interesting results, including a different strategy for*
*building living materials, as compared to the literature, via microfluidics and 3D*
*printing.”*

**Response:** We thank the reviewer for the comments.

**Reviewer 3:** *“However, I think this needs to be fleshed out quite a bit for a*
*publication at Nature Communications, when compared to current living material*
*papers published in this journal. The crux of the paper is based on bioprinting, yet*
*there does not seem to be a demonstration of the advantage to print these*
*structures.”*

**Response:** We thank the referee for this question and share some thoughts on this
topic below. The processing of micro-scale building blocks into functional materials is
an aspect which is crucial for the formation of macroscale materials and systems in
general. The problem in the area of living materials is not so much the processing

but so far the lack of building blocks suitable for printing and other processing
approaches. The main objective of our work is to solve this problem by developing
new bioink for bioprinting and to propose a novel method to control the cellular
microenvironment for such processes, and we demonstrate that our method can
greatly enhance the bioprocessing capability of the fabricated living materials.
Please see below further information regarding this point.

**Reviewer 3:** *“In general, the data does support the conclusions, but often I had to go
to the supplementary methods to understand the system. Moving parts of the
methods into the main text would be useful in some parts and I will highlight this
later. Importantly, the ecological terms used in this paper are sometimes not
accurate of the ecological literature and need to be corrected to not mislead the
reader.”*

**Response:** We thank the reviewer for the helpful suggestion. We have moved our
methods section in the revised manuscript and also corrected the terminology used.

**Reviewer 3:** *“Comment 1: When generating the core-shell hydrogel structures via
microfluidics the authors briefly mention in line 66-67 that this an aqueous two-phase
system. I presume this is between the gelatin and the and the
carboxymethylcellulose. Can the authors clarify this, and if so demonstrate this
behaves as an aqueous two-phase system as compared to liquid-liquid phase
separation papers in the literature? If I have understood this incorrectly, can the
authors add more detail why the two aqueous phases do not mix before the
hydrogelation of the shell.”*

**Response:** The reviewer is absolutely right, the formation of the core-shell structure
(before gelation of the shell) is the result of the phase behavior between gelMA and
carboxymethylcellulose. The phase separation behavior between gelMA (or gelatin),
with other biopolymers is recently receiving significant attention in biomaterial
science, which has been utilized to fabricate microgels with complex geometries (Y.
Xu et al., 10.1002/admi.202101071) and to create macroscopic pores in bioprinting
(G. Ying et al., 10.1002/adma.201805460). Following the reviewer’s useful advice,
we have now conducted a bulk experiment to demonstrate the phase separation of

gelMA and carboxymethylcellulose, in which we have found that their phase
separation is at least affected by the ionic strength (see attached below).

Figure R5. Phase separation of gelMA and CMC.

**Reviewer 3:** “Comment 2: In Fig. 1, the schematic looks like crosslinking is occurring
everywhere which is slightly confusing as from the text it should only be happening in
the shell. This is obviously not the authors intention.”

**Response:** We thank the reviewer for bringing up this issue. We have made some
alterations to the schematic Fig. 1a (please see comment 3 reviewer #2, and Figure
R1).

**Reviewer 3:** “Comment 3: Can the authors comment more on the environment of the
core as presumably this is a viscous liquid? It will be important for growth of the
microorganisms and cells as it will determine whether they are sedimented at the
bottom and growing, or indeed growing in 3D. A confocal z-stack would be useful to
show this compared to the x, y images shown in Fig. 3 for *E. coli* and HEK cells.”

**Response:** We thank the reviewer for the insightful question. We have performed
extra SEM characterization of the core-shell microgels to show that the core phase

was a liquid instead of a crosslinked hydrogel network. We also provided more
rheological data of the CMC solution (please see comment 6 reviewer #2). We did
carry out a confocal microscopic characterization of the HEK 293T spheroids with z-
stack images (please see the supplementary fig. 10), which shows that the structure
of the cellular spheroid is similar to that of previously reported spheroids that grow in
microgels, for example, S. Sart, et al., 10.1038/s41467-017-00475-x and H. Wang, et
al., 10.1002/admt.201800632. As for the cell culture environment, cells might
sediment in this viscous liquid, but we are not fully convinced that this would become
an important concern because even so they would still be largely growing in a 3D
space surrounded by the core material. Also the characteristic cellular structure of
the spheroids suggests that the cells indeed grow in 3D instead of a 2D environment.

**Reviewer 3:** *“Comment 4. Minor point, line 200, this is not bacterial community but a*
*bacterial population as it is a single species (E. coli). Minor points for the word*
*symbiosis, which is used in line 257 and lines 337-338. Symbiosis is often defined as*
*a long-term relationship between two organisms that often needs a host. My advice*
*would be to remove this term as it unlikely to apply to these microbial systems.”*

**Response:** We thank the reviewer for the helpful advice, and we have changed the
bacterial “communities” into “populations” and removed the word symbiosis.

**Reviewer 3:** *“Comment 5. For the relationship between C. vulagris and B. subtilis, it*
*is not clear how they degrade amoxicillin or methyl orange, or why you need both of*
*them to do this. Can this be clarified in the text. Moreover, reference 37 points to this*
*relationship between C vulgaris and B. subtilis for bioremediation of organic*
*contaminants – I could not find this in the reference (maybe I missed this). Can the*
*authors provide other experimental evidence for this or another reference that*
*explicitly shows this? In line 257, the authors write that the relationship between*
*these two microbes are a naturally occurring symbiosis. Can the authors provide*
*either a strong reference for this or experimental evidence for their interactions via*
*pairwise cultures (see reference: Mitri S, Foster KR (2013). Annual Review of*
*Genetics, 47:247–73, figure 5 for an explanation on how to do this).”*

**Response:** This is a good question, and we have added extra references regarding
the biodegradation of methyl orange (R. S. Masarbo, et al.,
10.1080/10889868.2018.1516612, and L. R. S. Pinheiro et al.,
doi.org/10.3390/su14031510) and amoxicillin (N. Chandel, et al.,
10.1016/j.scitotenv.2022.153895). Indeed from these references the degradation of
either chemical can be performed by only one microbe, in this case bacteria for
methyl orange and microalgae for amoxicillin. The reason why we used consortia
instead of single microbial populations was (1) to demonstrate the advantage of our
spatial division strategy using microgel bioprinting, which requires the involvement of
at least two microbes and (2) that we think the microalgae-bacteria consortium is a
type of rather common microbial consortium that has been intensively reported and
reviewed for bioremediation purpose (A. L. Gonçalves, et al.,
10.1016/j.algal.2016.11.008, S. S. Chan, et al., 10.1016/j.biortech.2021.126159, and
our previous work F. He, et al., 10.1002/sml.202104820). Therefore it is the
interaction between two species and the resultant collective synergistic effect that we
want to utilize to remediate the organic compounds. However we have now also
found that our results might not directly demonstrate it, namely the microgel spatial
segregation can improve such effect, and therefore we have added some extra
characterization to show this, in addition to our scaffold disintegration experiments
(reviewer#2, comment 11). Furthermore, we have removed the term “organic
contaminants” and “naturally occurring symbiosis” while adding more specific
references.

**Reviewer 3:** *“Comment 6. For the experiments related to Fig. 4C important controls*
*are missing such as homogenous PAM scaffolds (i.e. C. vulgaris and B. subtilis co-*
*encapsulated), and scaffolds just containing either C. vulgaris and B. subtilis, and*
*liquid culture (i.e., what has been similarly done for Fig. 4h). It is important to show*
*that heterogenous PAM scaffolds, i.e., segregating out C. vulgaris and B. subtilis is*
*beneficial for bioremediation. The authors allude to this later in the paragraph by a*
*degradation assay, however this does not directly measure bioremediation. This is*
*important to show, as if this cannot be shown, then the fabricated system is not*
*useful for this assay. One minor point is that it is not clear what is the morphology*
*and dimensions of 3 x 3 lattice shaped PAM scaffold. Can the authors include a*

*diagram or photograph of this. Was this 3 x 3 lattice used in the bioremediation*
*assay as well as the degradation assay”*

**Response:** Following the reviewer’s helpful suggestion, we now have added a new
result of bioremediation (please see comment 11, reviewer#2 and Figure R4B),
which shows that the spatial segregation of *C. vulgaris* and *B. subtilis* (the
heterogenous scaffold) could remove more methyl orange in 24 hours, compared to
homogeneous scaffolds and scaffolds with single microbial monocultures. The
morphology of the 3x3 lattice is similar across all the microbial assays regardless of
the types of microorganisms, which is shown in supplementary figure 13 and 16 (the
picture below), and the dimension is roughly 20x20 mm.

**Hand-extruded scaffold**

**Fermentation (1st batch, day 0)**

506

**Figure R6. Scaffolds used for microbial assay.**

**Reviewer 3:** “Comment 7. Line 299 the authors claim the *Escherichia coli* and
*Meyerozyma guilliermondii* form a competitive relationship. From Fig. 4h on co-
encapsulation, the production of 2-PEis lower than when segregating the species,
however, cell numbers are not measured, so competition between the species
cannot be inferred. Please confirm this from the literature or demonstrate via a liquid
pairwise competition assay mentioned in comment 5.”

**Response:** We thank the reviewer for the suggestion. We now have added a new
experimental result (please see below), which shows that *M. guilliermondii* is the
dominant species in this microbial consortium. When *E. coli* and *M. guilliermondii* are
inoculated simultaneously, the final ratio between *E. coli* and *M. guilliermondii* will be
around 1:9, regardless of the initial ratio.

**Figure R7.** Competition growth between *E. Coli* and *M. guilliermondii*.

**Reviewer 3:** “Comment 8. Line 304, the authors claim the endpoint production of 2-
PE at $14.23 \pm 1.12 \text{ mg mL}^{-1}$ is among the highest in the literature. Can references
and comparison values be provided.”

**Response:** We thank the reviewer for the advice. First please note that this
production of 2-PE, namely $14.23 \pm 1.12 \text{ mg/mL}$, is normalized by the hydrogel weight.
The non-normalized end point production is $6.21 \pm 0.73 \text{ mg/mL}$, which represents the
actual concentration we measured from the fermentation media. As for the
comparison with literature, here we refer to a few review papers: X. Qian, et al.,
[10.1080/07388551.2018.1530634](https://doi.org/10.1080/07388551.2018.1530634), and S. Mitri., et al., [10.3390/foods11010109](https://doi.org/10.3390/foods11010109).
However, after careful consideration, we have decided to remove this description
because that (1) the main focus of our results is to compare the production of 2-PE
from different patterns of cell spatial organization and therefore it is the difference
rather than the absolute value that matters and (2) this is not a traditional metabolic
engineering study and therefore the experimental setup was quite different from
those that specialize in fermentation studies, such as the size of fermentation
system, etc., and we now deem not totally appropriate making the comparison.

**Reviewer 3:** “Comment 9. The two microbial assays in Fig. 4 do not necessarily
directly demonstrate the benefit of the core-shell morphologies. This may be inferred

*that the microorganisms will grow poorly (and therefore may have difficult performing*
*bioremediation of 2-PE synthesis) from the E. coli results in Fig. 2B, however a direct*
*comparison would be useful i.e. showing how these microbes encapsulated in*
*homogenous microgels, printed, and cross-linked, perform compared to the core-*
*shell morphologies.”*

**Response:** This is an excellent suggestion. As discussed before, the reason we
used core-shell structure is to separate the material inherent mechanical property
from the cellular microenvironment, which in this case is the viscous core instead of
a crosslinked hydrogel network identical to the shell. We have already demonstrated
this strategy could mitigate cell leakage (see comment 7, reviewer#2) and now we
have a new experimental result to show another advantage of the core-shell strategy
(please see below Figure R8). We fabricated scaffolds from (1) core-shell microgels
with yeast cells and (2) non-core-shell microgels with yeast cells and tested their
performance in terms of anaerobic fermentation of glucose into ethanol in an
oxygen-free condition (F. He, et al., 10.1002/sml.202104820). The result shows that
strikingly, with the core-shell strategy, yeast can produce significantly higher ethanol
than the other group in first 12 hours. However, we also found that the end-point
ethanol production (20 hour) from the core-shell microgel scaffolds was slightly
lower, which we attributed to the cell leakage as again a much more pronounced cell
leakage problem was found for the non-core-shell microgels (Figure R9). Our new
result is in agreement with the finding of a recent paper where the author showed
that the stiffness of the hydrogel materials could slow down bacterial proliferation
and impact their metabolism (S. Bhusari, et al., 10.1002/adv.202106026) and hence
the delayed production of ethanol in non-core-shell microgel scaffolds. As a result,
we think our core-shell strategy is better suited to cell culture compared to non-core-
shell microgels, which adds a new advantage to our method.

Figure R8. Core-shell strategy promotes the anaerobic production of ethanol from yeast.

Figure R9. Yeast cell leakage after 20 hours of fermentation.

Reviewer 3: “Comment 10. The authors do not seem to clearly demonstrate the benefit of 3D printing for their microbial assays compared to moulding. i.e. the segregation of different microbial species is generated from randomly mixing the core-shell hydrogels rather than in the printing itself. The authors need to clearly demonstrate the benefit of 3D printing here to warrant this a useful technique. One suggestion would be to create lattices similar to what is seen in Fig. 1c and change the area of free space in-between the lattice to see if there is a difference in bioremediation or production 2-PE. Other suggestions are welcome to demonstrate the importance of 3D printing in this system.”

Response: We thank the reviewer for the insightful comment. Bioprinting, especially extrusion bioprinting, is a relatively well-established technique with well

characterized advantages over traditional biofabrication routes including molding, for
example the ability to build 3D structures in arbitrary patterns, a better control over
the spatial distribution of cell populations, etc., which has been demonstrated and
discussed by previous reports (Daly, et al., 10.1016/j.cell.2020.12.002; Zhang, et
al., 10.1038/s43586-021-00073-8). Our work proposes a fundamentally new way of
bioprinting that utilized discrete and tunable core-shell microgel building blocks as a
new type of bioink and we focused on the advantages of core-shell microgels in the
context of bioprinting, in terms of mitigation of cell leakage, suitability for cell culture,
and the ability to spatially segregate microbial consortia.

**Reviewer 3:** *“Comment 11. Another important point of the paper is confinement of*
*the microbes at the microscale is important for the bioactivities of the microbes (as*
*per line 31 in the abstract). Literature definitely points to this; however, this is not*
*explicitly demonstrated in this paper. However, it might be possible to show this. Is it*
*possible to create larger sized-core shell particles to show a difference in*
*bioactivities, i.e., mm-sized, or at least larger or smaller sized micron-sized*
*particles?”*

**Response:** This is a very good question. We have conducted new experiments
where two sizes of yeast-laden core-shell microgels were generated (see the results
below) and printed into scaffold for ethanol fermentation. The results show that the
ethanol production did not exhibit significant differences between two microgel sizes
with the same cell loading (please see more information in the revised manuscript).
The reason can be that due to the microporosity of the material plus the short
diffusion length from the cells to the medium through the microgel shell, the material
does not pose a noticeable diffusional barrier for the metabolite and hence no
significant differences in performance. Since our work is based on droplet
microfluidics, and the printed filaments are also sub-millimeter (please refer to fig.2 D
and F), it would have been challenging to make mm-sized core-shell microgels for
bioprinting. Therefore we have changed the description of “microscale” into a more
specific term “microgels”.

Figure R10. Performance of scaffolds assembled from differently sized microgels.

**Reviewer 4:** *“This work reports an approach for constructing living materials via a*
*common extrusion bioprinting routine. In this approach, jammed core-shell microgels*
*with bacteria or mammalian cells in the aqueous cores are interparticle annealed to*
*give covalently stabilized functional scaffolds with controlled microporosity that*
*enhances the mass transfer of nutrients and metabolites. In general, the topic is*
*interesting, and the manuscript is well-written.”*

**Response:** We thank the reviewer for the kind comment of our work.

**Reviewer 4:** *“However, it seems that the advantage of the proposed strategy in this*
*work is not so obvious compared to those in the previous reports (e.g. Highley, C.*
*B.et al., Adv. Sci. 6, (2019); Chai, N. et al., COMPOSITES PART B-ENGINEERING,*
*2021,109100). As such, I recommend a major revision before its acceptance in the*
*high-quality journal of Nature Communications. Other specific comments are listed*
*as below.”*

**Response:** This is an insightful comment and in the revised paper we have cited the
references that the referee suggests (the first one was already included in the
original submission). The essence of our work is to introduce a different idea from
traditional bioprinting routines. Via a core-shell microgel strategy, the cell culture
environment can be decoupled from the material processed by 3D printing. To fully
showcase the advantages of this core-shell approach, we show that our method
could reduce cell leakage to the media and provides a more suitable environment for
cell culture in comparison with a simple non-core-shell counterpart. Furthermore,
through the spatial segregation strategy, our method enhances the interactions
between microbial consortia. We believe these results are able to demonstrate the
advantages of our method, and more importantly, can bring new perspectives to the
bioprinting community. Please see more detailed discuss in the response.

**Reviewer 4:** *“1. As the authors mentioned in the manuscript, several works related*
*to jammed microgel printing have been reported (Highley, C. B. et al., Adv. Sci. 6,*
*(2019)). Even the similar method for core-shell microgel printing has been proposed*
*(Chai, N. et al., COMPOSITESPART B-ENGINEERING, 2021, 109100). So, the*

*authors should provide more evidence or data to prove the significant advantages of*
*this manuscript compared to the previous reports.”*

**Response:** We thank the reviewer for the comment and we have included these
references. First of all, the key novelty in our work is that we introduced a new way of
printing cell-laden structures that can separate the cell microenvironments from the
material for bioprinting. Moreover, we combine droplet microfluidics and microgel
extrusion printing not only to immobilize cells, but also to segregate cell populations
spatially to increase the interactions between different cell populations and hence
enhanced bioactivities, which we demonstrated using two microbial consortia. To the
best of our knowledge, this is a new kind of application scenario in the field of
functional living materials and hence the major innovation of our work.

Regarding the advantages of our work, technique-wise, although extrusion printing
jammed microgels was reported initially by the Burdick group in 2019 (C. B. Highley,
et al., 10.1002/adv.201801076), they did not use the printed structures for any
specific functions but merely a seminal demonstration; however in our paper, we
emphasis on the functionality of the printed microbe-laden scaffolds and how we
could use droplet microfluidics and extrusion bioprinting to increase the
bioprocessing capacity of such structures. Therefore, we believe that our work
represents a technical advance in the field of functional living materials, if not in the
field of extrusion printing microgels. Concerning the advantage over the second work
mentioned (N. Chai, et al., 10.1016/j.compositesb.2021.109100), the authors used a
different method, namely, that they did not directly print jammed microgels but a
microgel suspension in a secondary material (in this case the cores-shell microgels
in silMA), with its invention predating the former technique by at least two years
(please refer to this paper published in 2017: T. Kamperman, et
al., 10.1002/adhm.201600913) and a number of similar works recently: M. Xie, et
al., 10.1038/s41467-022-30997-y; Y. Fang, et al., 10.1002/adfm.202109810; Q.
Feng, et al., 10.1021/acsami.2c01295. Therefore we are not fully convinced that it is
ideal to compare our work with this one based on that (1) a review paper in Nature
Review Materials (A. C. Daly, et al., 10.1038/s41578-019-0148-6) has put jammed
microgels (the “granular microgels”) and microgel suspensions in a secondary
material (the “HMP composite”) as different subcategories of hydrogel microparticle

systems (please refer to figure 1 in this review). (2) The two material systems carry
very distinctive physical properties: while the printability of jammed microgels arise
from the physical interactions between microgels in a packed state and therefore it
decouples polymer chemistry from printability (C. B. Highley, et
al., 10.1002/adv.201801076), the mechanical property of the latter system is largely
resulted from the secondary material that often needs rheological tuning in order to
print (A. C. Daly, et al., 10.1038/s41578-019-0148-6), which is a disadvantage in our
opinion. Moreover, the jammed microgels possess an additional advantage of
microporosity over the other system. (3) The applications were different. The
mentioned work centers on mammalian cell-based biofabrication and regenerative
medicine whereas ours on functional microbes-laden scaffolds for bioprocessing,
which we think are rather dissimilar areas of research. (4) Regarding the core-shell
microgels, we intended to utilize this strategy to decouple the inherent microgel
mechanical environment from cell culture environment, namely the shell and the core
phase respectively, and we did observe distinctive morphologies of bacterial
microcolonies in core-shell and homogeneous microgels as well as different cell
leakage. Moreover, now we have provided a new result to show that the core-shell
strategy further augments the bioprocessing capacity of fabricated living materials. It
is reported recently that a more elastic environment would slow down the
proliferation of bacterial populations (S. Bhusari, et al., 10.1002/adv.202106026)
embedded in hydrogels and therefore we believe the core-shell structure is more
suitable for cell culture amid a satisfactory printing behavior. However, in this
mentioned paper, the reason for using core-shell microgels seems unclear to us. In
conclusion, we believe our work represents a significant advancement compared
with these two works.

**Reviewer 4:** *“2. In regard to the characterization of the microgels, the authors should*
*provide more data to demonstrate that the cores are liquid rather than solid. In*
*addition, in Fig S2, it seems the compartmentalization of the microgels are not*
*obvious. The authors should provide images with higher definition.”*

**Response:** This is a very good suggestion. We have now added a set of new
rheology characterization of the core material. Please refer to comment 6,
reviewer#2. Regarding the original Fig S2, after careful consideration, we have

decided to remove the fluorescence image of the compartmentalization of the cells in
the different phases of the core-shell microgels, although we find it potentially
another interesting way to spatially segregate microbial consortia in the first place.

**Reviewer 4:** “3. In the paper, the authors claimed that the dual covalent strategy is
reversible, i.e., droplets can be first cured by blue light radiation and then annealed
enzymatically. However, the enzymatic reaction needs overnight incubation at room
temperature. Such a low reaction is not suitable for the fast-printing process. It
seems like a limitation of the proposed method.”

**Response:** We thank the reviewer for the comment. In this reversed strategy,
namely, blue light radiation to cure the droplets and then enzymatic annealing, the
scaffolds were annealed for 30 mins instead of overnight for rheological
characterization (please refer to the methods section, *rheology*). In our experiences,
using this strategy, the scaffold could be annealed under 1 hour (similar to that
observed in this *Nature Materials* paper, where the authors also used enzymatic
annealing: D. R. Griffin, et al., 10.1038/nmat4294), but we found this would lead to
weaker mechanical properties of the scaffold (please refer to supporting information,
Supplementary Fig. 5F) compared to the main strategy used in our work.

The reason why we incubated the droplets overnight was to render them well-cured
(please see comment 4, reviewer #1) and meanwhile the microbes could pre-
proliferate into microbial populations before bioprinting, which we deem conducive to
the following bioprocessing process. Though the reversed strategy might not be
perfectly suited to our demonstrated applications, namely microbial bioprocessing,
we think it will find its better utility in mammalian cells-based applications, such as
tissue engineering and biofabrication.

**Reviewer 4:** “4. As the dual-reaction process is crucial for the formation of printed
scaffolds, the authors should provide more data to prove the existence of the
enzymatic and photo-initiated bonds in the microgels.”

**Response:** We thank the reviewer for the question. For the photo-initiated bonds, we
in our original manuscript included an NMR characterization in the supporting

information (please refer to Supplementary Fig. 2) that shows the double-bond
functionalization of the gelatin. Moreover, the rheology experiments show that the
mechanical strengthen of the scaffold, regardless of the order of the dual co-valent
reactions, increases after photo-initiated/enzymatic annealing process, suggestive of
the efficacy of the reactions. Lastly, to directly show the enzymatic reaction, we
incubated at room temperature for 2 hours core-shell droplets (1) with 10 U/mL
transglutaminases and 2) without enzymes and used PFO to demulsify both
emulsions. Results (below) show that droplets without the enzyme completely
merged whilst the other group only witnessed few droplets merging.

**Figure 11. Demonstration of the enzymatic reaction.**

**Reviewer 4:** “5. The authors have fully demonstrated the application of bacteria-
laden scaffolds in bioremediate organic contaminants. What are the application
scenarios for cell-laden scaffolds?”

**Response:** We thank the reviewer for the comment. The protein nature of the
materials is better suited to culture mammalian cells (S. R. Caliarì and J. A.
Burdick, 10.1038/nmeth.3839) and the reversed dual-network strategy is faster and
mechanically softer than the current strategy, which we consider more suitable for
biofabrication (please refer to the comment 3). The reason we used gelMA is to

maximize the material biocompatibility as we intended to prioritize biological
applications over the specific material selected. Therefore, we are also working on
using this strategy to develop new methods for biofabrication, which is beyond the
scope of the current study.

**Reviewer 4:** *“6. Fig 3A, it seems that the fluorescence intensity in microgels is*
*homogeneous at 0 h, why?”*

**Response:** We thank the reviewer for pointing this out. The *E. coli*. cells used in this
experiment were genetically engineered with a bacterial quorum sensing module that
responded to the presence of *N*-Acyl homoserine lactone (AHL) molecules and
reported via the expression of eGFP, the same to the ones we used previously (F.
He, et al., 10.1002/sml.202104820 and S. Zhao, et al., 10.1016/j.snb.2021.129648).
The fluorescence image was therefore homogenous because it was taken right after
the resuspension of bacteria-laden microgels into the AHL-containing culture
medium and hence very lower eGFP expression. We have added that information to
the revised manuscript.

**Reviewer 4:** *“7. What is the exact dimension of the cellular spheroid in Fig S11? The*
*dimensions are inferred about 100 micrometers from other figures. Generally,*
*cellular spheroids at such dimensions seldom show necrotic cores. In addition, the*
*dead cells randomly distributed in the cellular spheroid. Do these phenomena*
*indicate that the presence of microgels decrease the cell viability? The authors*
*should provide more control data to confirm the doubt.”*

**Response:** We thank the reviewer for the question. After careful checking, the
scales bars represent 50 micrometers in Fig.S11 in the original manuscript. The
reason for the occurrence of the necrotic cores can be multifaceted, such as the
physical/biophysical barrier of the stiffer shell, insufficient nutrients transfer due to
nanoporosity etc., which we think it will be better discussed in a more specialized
paper. As a result, we have removed the description and the data regarding the
necrotic cores of the HEK 293t spheroids so that it would not further mislead
readers.

REVIEWERS' COMMENTS

Reviewer #1 (Remarks to the Author):

In this manuscript, the authors propose a new method to address this challenge. In this work, we explore the unique applicability of cell-laden core-shell microgels at the interface of bioprinting and functional living materials. I think the authors have addressed the reviewers' questions. Here, I want to suggest a few production problems need to be considered before publishing:

1. The scale bar in some of the figures should be easy to read (e.g., Figure 2f).
2. Please modify the font size in all the figures (e.g., Figure 1b, Figure 2a, Figure 3d, Figure 4c&d).
3. There are writing inconsistencies etc. present that should be carefully examined by the authors.
4. There are two pages of original data in the end of the PDF, I think these materials should be listed in the supporting part.

Reviewer #2 (Remarks to the Author):

The authors addressed all my comments nicely. They greatly improved the quality of the manuscript, both scientifically and visually. I do not have further questions about this work but do have two suggestions.

1. Please define the lower limit of detection while presenting the plating/CFU data.
2. Could the authors use a few sentences to discuss other scenarios (besides incubating in liquid medium where cell leakage is a problem) in which their technology could be useful for? Could this be used to make a standalone device for some specific applications in non-liquid environments? What extra features would be added?

Reviewer #3 (Remarks to the Author):

The authors have addressed my comments as well as providing excellent additional experimental results. In terms of justification of using bioprinting and the focus of the paper (comment 1 and 10): 'bioprinting that utilized discrete and tuneable core-shell microgel building blocks as a new type of bioink and we focused on the advantages of core-shell microgels in the 596 context of bioprinting, in terms of mitigation of cell leakage, suitability for cell culture', this has been strengthened well in the main text.

I appreciate that some experiments could not be performed (comment 11), i.e., making mm sized core-shell particles to compare if the microscale length scales are indeed important for these processes.

The extra experiments and generated figure (Fig .4) really strengthen the justification of a core-shell morphology vs. homogeneous morphology. One small minor comment in line 277-279, it could be explained better why the ethanol production was higher at the end point measurement for hydrogel microgels vs core-shell microgels like explained in the response to reviewer comment: 'However, we also found that the end-point ethanol production (20 hour) from the core-shell microgel scaffolds was slightly lower, which we attributed to the cell leakage as again a much more pronounced cell leakage problem was found for the non-core-shell microgels (Figure R9). Our new result is in agreement with the finding of a recent paper where the author showed that the stiffness of the hydrogel materials could slow down bacterial proliferation and impact their metabolism (S. Bhusari, et al., 10.1002/adv.202106026) and hence the delayed production of ethanol in non-core-shell microgel scaffolds. As a result, we think our core-shell strategy is better suited to cell culture compared to non-core shell microgels, which adds a new advantage to our method.'

Taken everything into account, I would recommend this paper for publication at Nature Communications.

Reviewer #4 (Remarks to the Author):

The authors have addressed all my comments. As such, I recommend its acceptance at this stage.

Bioprinting microporous functional living materials from protein-based core-shell microgels

Point-by-point responses to reviewers' comments

Reviewer #1

"In this manuscript, the authors propose a new method to address this challenge. In this work, we explore the unique applicability of cell-laden core-shell microgels at the interface of bioprinting and functional living materials. I think the authors have addressed the reviewers' questions."

Response: we thank the reviewer for their comments and time taken to review our work.

"Here, I want to suggest a few production problems need to be considered before publishing:

- 1. The scale bar in some of the figures should be easy to read (e.g., Figure 2f).*
- 2. Please modify the font size in all the figures (e.g., Figure 1b, Figure 2a, Figure 3d, Figure 4c&d).*
- 3. There are writing inconsistencies etc. present that should be carefully examined by the authors.*
- 4. There are two pages of original data in the end of the PDF, I think these materials should be listed in the supporting part."*

Response: we thank the reviewer for these useful suggestions. We have made all the alterations and corrections accordingly.

Reviewer #2

“The authors addressed all my comments nicely. They greatly improved the quality of the manuscript, both scientifically and visually. I do not have further questions about this work but do have two suggestions.”

Response: we are grateful for the reviewer’s kind comments and the time invested to review our work.

“1. Please define the lower limit of detection while presenting the plating/CFU data.”

Response: we have revised and added more information in the supplementary information.

“2. Could the authors use a few sentences to discuss other scenarios (besides incubating in liquid medium where cell leakage is a problem) in which their technology could be useful for? Could this be used to make a standalone device for some specific applications in non-liquid environments? What extra features would be added?”

Response: this is a very good suggestion. In fact there are some good examples where 3D printing microbes can be used in a non-liquid environments, for example, for sensing applications (DOI: 10.1073/pnas.1618307114). Besides this, a very recent publication has shown even the cell leakage problem itself can be a new strategy for controlled microbial cell release, by manipulating the elastic property of the matrix.

Reviewer #3

“The authors have addressed my comments as well as providing excellent additional experimental results. In terms of justification of using bioprinting and the focus of the paper (comment 1 and 10): ‘bioprinting that utilized discrete and tuneable core-shell microgel building blocks as a new type of bioink and we focused on the advantages of core-shell microgels in the 596 context of bioprinting, in terms of mitigation of cell leakage, suitability for cell culture’, this has been strengthened well in the main text.”

Response: we thank the reviewer for their helpful feedback and comments.

“I appreciate that some experiments could not be performed (comment 11), i.e., making mm sized core-shell particles to compare if the microscale length scales are indeed important for these processes.”

Response: we thank the reviewer for their appreciation of not performing this experiment. Indeed how the microscopic length scale of the building blocks affects the macroscopic performance of the material is a rather interesting topic that is not yet sufficiently studied, which we will be investigating in our future work.

“The extra experiments and generated figure (Fig .4) really strengthen the justification of a core-shell morphology vs. homogeneous morphology. One small minor comment in line 277-279, it could be explained better why the ethanol production was higher at the end point measurement for hydrogel microgels vs core-shell microgels like explained in the response to reviewer comment: ‘However, we also found that the end-point ethanol production (20 hour) from the core-shell microgel scaffolds was slightly lower, which we attributed to the cell leakage as again a much more pronounced cell leakage problem was found for the non-core-shell microgels (Figure R9). Our new result is in agreement with the finding of a recent paper where the author showed that the stiffness of the hydrogel materials could slow down bacterial proliferation and impact their metabolism (S. Bhusari, et al., 10.1002/advs.202106026) and hence the delayed production of ethanol in non-core-shell microgel scaffolds.

As a result, we think our core-shell strategy is better suited to cell culture compared to non-core shell microgels, which adds a new advantage to our method.’ ”

Response: we thank the reviewer for this suggestion and we have made some alterations to the description in the manuscript.

“Taken everything into account, I would recommend this paper for publication at Nature Communications.”

Response: we are very grateful to the reviewer for their time spent reviewing our work.

Reviewer #4

“The authors have addressed all my comments. As such, I recommend its acceptance at this stage.”

Response: we thank the reviewer for their comments and the time for the review of our manuscript.